# DIVERSIFYING SPURIOUS SUBGRAPHS FOR GRAPH OUT-OF-DISTRIBUTION GENERALIZATION

## ABSTRACT

Environment augmentation methods have gained some success in overcoming the out-of-distribution (OOD) generalization challenge in Graph Neural Networks (GNNs). Yet, there exists a challenging *trade-off* in the augmentation: On one hand, it requires the generated graphs as *diverse* as possible to extrapolate to unseen environments. On the other hand, it requires the generated graphs to preserve the invariant substructures causally related to the targets. Existing approaches have proposed various environment augmentation strategies to enrich spurious patterns for OOD generalization. However, we argue that these methods remain limited in *diversity and precision* of the generated environments for two reasons: i) the deterministic nature of the graph composition strategy used for environment augmentation may limit the diversity of the generated environments, and ii) the presence of spurious correlations may lead to the exclusion of invariant subgraphs and reduce the precision of the generated environments. To address this trade-off, we propose a novel paradigm that accurately identifies spurious subgraphs, and an environment augmentation strategy called *spurious subgraph diversification*, which extrapolates to maximally diversified spurious subgraphs by randomizing the spurious subgraph generation, while preserving the invariant substructures. Our method is theoretically sound and demonstrates strong empirical performance on both synthetic and real-world datasets, outperforming the second-best method by up to 24.19% across 17 baseline methods, underscoring its superiority in graph OOD generalization.

## 1 INTRODUCTION

GNNs (Kipf & Welling, 2017; Xu et al., 2019; Veličković et al., 2017) have demonstrated exceptional performance in learning from graph-structured data across diverse fields (Qiu et al., 2018; Wu et al., 2022b; Yu et al., 2018; Zhang et al., 2022c). However, it generally assumes that the training and test graphs are independently drawn from the identical distribution, which often fails in many real-world graph applications (Hu et al., 2020; Huang et al., 2021; Ji et al., 2022; Koh et al., 2021). Such distribution shifts can drastically undermine the generalization capabilities of GNNs, hindering their applicability in practical situations.

To address the OOD generalization challenge, recent studies have explored data augmentation methods across various domains (Shorten & Khoshgoftaar, 2019; Yao et al., 2022; Park et al., 2022; Kong et al., 2022; Han et al., 2022). In the context of graph data, graph data augmentation (GDA) methods such as DropEdge (Rong et al., 2019), FLAG (Kong et al., 2022), and M-Mixup (Wang et al., 2021) perturb graph features or structures to enlarge the training distribution to facilitate generalization on unseen environments. However, GDA methods are prone to perturbing stable features or patterns that are critical for the predictive task, potentially limiting their effectiveness for OOD generalization. Inspired by causality (Peters et al., 2016) and invariant learning (Arjovsky et al., 2020; Krueger et al., 2021), recent studies have proposed environment augmentation methods (Wu et al., 2022c; Liu et al., 2022; Sui et al., 2023; Li et al., 2024) that generate environment-sensitive substructures while preserving invariant patterns. The goal is to capture stable features by learning equipredictive encoders across different environments. A key condition for the success of these methods lies in the *diversity and precision* of the generated environments, represented by spurious subgraph patterns that capture rich environmental variations (Sui et al., 2023; Li et al., 2024). Notably, DIR (Wu et al., 2022c) and GREA (Liu et al., 2022) perform input-level and latent-level augmentation

respectively to create diverse environments. However, these methods rely on interpolation paradigms, which limit the diversity of the generated environments. In addition, some other studies explore environment extrapolation to generate rich spurious patterns for unseen environments, utilizing adversarial augmentation (Sui et al., 2023), and linear extrapolation in graph space (Li et al., 2024).

Despite these advancements, we argue that the *diversity and precision* of the generated environments in existing methods remain limited for two reasons. First, most of these methods adopt closed-form combination strategies (See Appendix G for details) to composite invariant and spurious subgraphs (Wu et al., 2022c; Liu et al., 2022; Sui et al., 2023; Li et al., 2024). This paradigm inherently constrain the diversity of the generated spurious patterns due to their deterministic nature. Second, the presence of spurious correlations can lead to the incorrect exclusion of (a portion of) invariant subgraphs during environment augmentation, thereby reducing the precision of the generated environments. This raises our research question:

*How can we generate high-quality spurious patterns in terms of diversity and precision to enhance OOD generalization for graph data?*

To address this challenge, we propose a novel learning framework, which differs from previous environment augmentation methods. Specifically, our approach builds on the theoretical results in Prop. 2 that, under mild assumptions (Assumption 1 and 2), edges in the invariant subgraphs tend to exhibit higher predicted probabilities in the learnable data transformation compared to spurious edges. Therefore a) *to identify edges from spurious subgraphs accurately*, we utilize the bottom $K\%$ of edges with the lowest predicted probabilities as estimated spurious edges, and the subsequent diversification process operates exclusively on these edges, ensuring the invariant subgraphs unaffected. b) *To address the diversity issue of generated environments*, we propose *spurious subgraph diversification*, an effective environment extrapolation strategy that randomizes the generation of spurious subgraphs to encourage diversity of the generated environments. We additionally propose a graph size constraint to prune spurious edges and reduce the candidate space for spurious subgraphs to be generated, achieving more effective environment extrapolation. Our contributions can be summarized as follows:

- **Novel framework.** We propose iSSD, a novel learning framework that: i) **i**dentifies spurious subgraphs accurately, ensuring that invariant patterns remain unaffected during augmentation; and ii) randomizes the generation of spurious subgraphs within a reduced search space with **S**purious **S**ubgraph **D**iversification for effective environment extrapolation and OOD generalization.

- **Theoretical guarantee.** We provide theoretical analysis showing that a) The proposed graph size constraint provably enhances OOD generalization by tightening the OOD generalization bound (Theorem 3.1); b) Spurious subgraph diversification provably enhances OOD generalization by identifying the true invariant subgraphs (Theorem 4.2).

- **Strong empirical performance.** We conduct experiments on both synthetic datasets and real-world datasets, compare against 17 baselines, our method outperform the second-best method by up to $24.19\%$, highlighting the superiority of our proposed method.

## 2 PRELIMINARY

**Notation.** Throughout this work, an undirected graph $G$ with $n$ nodes and $m$ edges is denoted by $G := \{\mathcal{V}, \mathcal{E}\}$, where $\mathcal{V}$ is the node set and $\mathcal{E}$ denotes the edge set. $G$ is also represented by the adjacency matrix $\mathbf{A}$ and node feature matrix $\mathbf{X} \in \mathbb{R}^{n \times D}$ with $D$ feature dimensions. We use $G_c$ and $G_s$ to denote invariant subgraph and spurious subgraph. $\widehat{G}_c$ and $\widehat{G}_s$ denote the estimated invariant and spurious subgraph. $t(\cdot)$ refers to a (learnable) data augmentation function, $\widetilde{G} \sim t(G)$ represents $\widetilde{G}$ is sampled from $t(G)$, for simplicity, we may use $t(G)$ to denote a graph sampled from $t(G)$, e.g., $H(Y \mid t(G))$. We use $[K] := \{1, 2, \cdots, K\}$ to denote a index set, $\mathbf{w}$ to denote a vector, and $\mathbf{W}$ as a matrix respectively. Finally, a random variable is denoted as $W$, a set is denoted using $\mathcal{W}$. A more complete set of notations is presented in Appendix A.

**OOD Generalization.** In this work we consider the problem of graph classification under various forms of distribution shifts in hidden environments. Given a set of graph datasets $\mathcal{G} = \{G^e\}_{e \in \mathcal{E}_{\text{tr}} \subseteq \mathcal{E}_{\text{all}}}$, a GNN model $f = \rho \circ h$, comprises an encoder $h \colon \mathbb{R}^{n \times n} \times \mathbb{R}^{n \times D} \to \mathbb{R}^F$ that learns a representation $\mathbf{h}_G$ for each graph $G$, followed by a downstream classifier $\rho \colon \mathbb{R}^F \to \mathbb{Y}$ to predict the label

$\widehat{Y}_G = \rho(\mathbf{h}_G)$. In addition, a learnable data transformation function $t\colon \mathbb{R}^{n \times n} \to \mathbb{R}^{n \times n}$ is employed to generate a graph with only structural modifications. The objective of OOD generalization is to learn an optimal composite function $f \circ t$ that can simultaneously learn diverse and useful representations from ERM and identify invariant subgraph $G_c$ to improve OOD generalizability.

**Assumption 1.** Given a graph $G \in \mathcal{G}$, there exists a stable subgraph $G_c$ for every class label $y$, satisfying: a) $\forall e, e' \in \mathcal{E}_{tr}, P^e(Y \mid G_c) = P^{e'}(Y \mid G_c)$; b) The target $Y$ can be expressed as $Y = f^*(G_c) + \epsilon$, where $\epsilon \perp\!\!\!\perp G$ represents random noise, and $\perp\!\!\!\perp$ indicates statistical independence.

Assumption 1 has been widely adopted in previous graph invariant learning literature (Yang et al., 2022; Li et al., 2022b;a; Wu et al., 2022a;c; Liu et al., 2022; Chen et al., 2022). This assumption posits that a subgraph pattern $G_c$ is not only stably associated with the target label $Y$ across different environments but also retains sufficient predictive power for accurately determining $Y$.

**Assumption 2.** The mutual information between the invariant subgraph $G_c$ and the target label $Y$ is greater than that between the spurious subgraph $G_s$ and $Y$, i.e., $I(G_c; Y) > I(G_s; Y)$.

Assumption 2 naturally follows from Assumption 1, implying that $G_c$ is both more stable and more predictive of $Y$ compared to $G_s$. Assumption 2 is also consistent with many real-world applications, such as molecular property prediction, where specific motifs within a molecule are crucial for determining key properties like solubility, reactivity, or toxicity. These motifs, analogous to $G_c$, exhibit strong and stable relationships with the target properties across different environments.

## 3 THEORETICAL MOTIVATION

Our goal is to learn the conditional probability $\mathbb{P}(Y \mid G_c)$ for OOD generalization, which can be expressed as follows:

$$\mathbb{P}(Y \mid G_c) = \sum_{G_s} \mathbb{P}(Y, G_s \mid G_c) = \sum_{G_s} \mathbb{P}(Y \mid G_c, G_s) \mathbb{P}(G_s) = \mathbb{E}_{G_s}[\mathbb{P}(Y \mid G_c, G_s)]. \quad (1)$$

The second equality in Eqn. 1 requires that $G_s \perp\!\!\!\perp G_c$, which is achievable by diversifying $G_s$ (Sec. 4), thus making it uninformative to $G_c$. Eqn. 1 implies that to accurately learn $\mathbb{P}(Y \mid G_c)$, we need to marginalize over all possible patterns from spurious subgraphs $G_s$, which also aligns with effective environment extrapolation tackled in previous studies (Li et al., 2024; Sui et al., 2023). Consequently, obtaining a diverse set of spurious patterns is essential for learning $\mathbb{P}(Y \mid G_c)$ accurately. To address this challenge, we first propose *learnable data augmentation*, which is necessary and widely adopted in previous graph invariant learning methods (Wu et al., 2022c; Sui et al., 2023; Chen et al., 2022; Miao et al., 2022) for learning spurious subgraph patterns, as defined in below:

**Definition 1.** (*Learnable Data Transformation*) Consider a graph $G \in \mathcal{G}$ and let $\mathcal{T}_\theta$ denote a family of data transformations parameterized by $\theta \in \Theta$. A specific transformation $t(\cdot) \in \mathcal{T}_\theta$ is referred to as a learnable data transformation, which produces a modified structural representation of the graph $G$, while ensuring that $sim(t(G), G) < \delta$, where $sim(\cdot, \cdot)$ denotes the similarity function and $\delta$ is a positive scalar to avoid trivial solutions.

Next, we define *spurious subgraph diversification, which will utilized in the following section.*

**Definition 2.** (*Spurious Subgraph Diversification*) Let $\widetilde{G}$ be the sampled structural view from $t(G)$, $\widetilde{G}$ consists of invariant subgraph $\widetilde{G}_c$ and spurious subgraph $\widetilde{G}_s$, i.e., $\widetilde{G} = \widetilde{G}_c \cup \widetilde{G}_s$, *spurious subgraph diversification* aims to achieve the following goal:

$$\max_\theta \ H(\widetilde{G}_s), \text{ s.t., } \min \ H(\widetilde{G}_c). \quad (2)$$

Building upon Def. 1, we propose a lower bound of the mutual information $I(G; Y)$ as the objective:

**Theorem 3.1.** *Let* $\widetilde{G} \sim t(G)$ *be a structural view sampled from* $t(G)$*, the following inequality holds:*

$$I(G; Y) \geq -\lambda H(Y \mid f(\widetilde{G})) + (1 - \lambda) H(\widetilde{G}) - (1 - \lambda) H(\widetilde{G} \mid Y), \forall \lambda \in (0, 1). \quad (3)$$

We prove Theorem 3.1 in Appendix E.1. The lower bound in Eqn. 3 can be viewed as a reflection of Eqn. 1 from an information-theoretic perspective. Minimizing $H(Y \mid \widetilde{G})$ is equivalent to maximizing $\mathbb{P}(Y \mid G_c, G_s)$, and maximizing the Shannon entropy $H(\widetilde{G})$ improves the diversity of the sampled subgraph from $t(G)$. However, directly diversifying $H(\widetilde{G})$ may hurt the OOD generalization ability as $G_c$ is also diversified. To strike a balance between OOD generalizability and the enrichment of spurious patterns, we propose *spurious subgraph diversification*, as defined in Def. 2.

To highlight the advantages of *Spurious Subgraph Diversification*, we present the following proposition:

**Proposition 1.** *Spurious subgraph diversification (Eqn. 2) simultaneously enhances OOD generalizability through preserving $G_c$, and facilitates diversification of spurious subgraphs.*

Prop. 1 demonstrates that by employing spurious subgraph diversification, $t(\cdot)$ is able to generate enriched spurious patterns while preserving $G_c$, thereby better estimating $\mathbb{P}(Y \mid G_c)$. Next we will introduce our proposed method, building upon the theoretical motivations.

## 4 PROPOSED FRAMEWORK

In this section, we present our proposed framework iSSD, which is grounded in the theoretical motivations discussed previously.

**Learnable Data Transformation.** We begin with the introduction of the learnable data transformation function. We adopt learnable edge dropping as the class of data transformation $\mathcal{T}_\theta$. The rationale behind employing edge dropping stems from our presumed data generating process that there exists an invariant subgraph $G_c$ that is causally related to the target label $Y$. By employing a learnable edge dropping function $t(\cdot)$, $t(\cdot)$ will tend to retain $G_c$, and discard $G_s$, as implied in Proposition 2. Following previous studies (Miao et al., 2022; Luo et al., 2020; Ying et al., 2019), we model each edge $e_{ij} \sim Bernoulli(p_{ij})$ independently which is parameterized by $p_{ij}$. The probability of the graph $G$ is factorized over all the edges, i.e., $P(G) = \prod_{e_{ij} \in \mathcal{E}} p_{ij}$. To parameterize $\mathcal{T}_\theta$, we employ a GNN model to derive the node representation for each node $v$, followed by an MLP to obtain the logits $w_{ij}$ as following:

$$
\begin{aligned}
\mathbf{h}_v &= \text{GNN}(v \mid G), \ v \in \mathcal{V}, \\
w_{ij} &= \sigma\left(\text{MLP}\left(\mathbf{h}_i, \mathbf{h}_j, \mathbf{h}_i \| \mathbf{h}_j\right)\right), e_{ij} \in \mathcal{E},
\end{aligned}
\tag{4}
$$

here $\|$ denotes the concatenation operator. To ensure the sampling process from $w_{ij}$ is differentiable and facilitate gradient-based optimization, we leverage the Gumbel-Softmax reparameterization trick (Maddison et al., 2016) and Straight-Through (ST) estimator (Bengio et al., 2013), which is applied as follows:

$$
\begin{aligned}
p_{ij} &= \sigma\left((\log \epsilon - \log(1 - \epsilon) + \omega_{ij})/\tau\right), \epsilon \sim \mathcal{U}(0, 1), \\
\tilde{\mathbf{A}}_{ij} &= 1 - \text{sg}(p_{ij}) + p_{ij},
\end{aligned}
\tag{5}
$$

here $\tilde{\mathbf{A}}$ denotes the sampled adjacency matrix, $\tau$ is the temperature, $sg(\cdot)$ denotes the stop-gradient operator, and $\mathcal{U}(0, 1)$ denotes the uniform distribution. In Eqn. 5, we first calculate the parameter $p_{ij}$ for each edge $e_{ij}$, we then sample edges according to $Bernoulli(p_{ij})$, for the sampled edge $e_{ij}$, the ST trick (second line of Eqn. 5) ensures $\tilde{\mathbf{A}}_{ij}$ remain binary yet differentiable for gradient-based optimization. We then take $t(G)$ and $Y$ as inputs to the GNN model $f(\cdot)$ to compute the cross-entropy loss $\mathcal{L}_{GT}$ as follows:

$$
\mathcal{L}_{GT} = -\mathbb{E}_{\mathcal{G}} \sum_{k \in \mathcal{C}} Y_k \log\left(f(t(G))_k\right),
\tag{6}
$$

where $Y_k$ denotes the ground-truth label $k$ for graph $G$, and $f(t(G))_k$ is the predicted probability for class $k$ of graph $G$.

**Graph size constraint.** To enforce the constraint $sim(t(G), G) < \delta$, which helps avoid trivial solutions, we introduce a regularization term $\mathcal{L}_e$ which encourages a graph size distinction between $t(G)$ and $G$:

$$
\mathcal{L}_e = \mathbb{E}_{\mathcal{G}}\left(\frac{\sum_{(i,j) \in \mathcal{E}} \tilde{\mathbf{A}}_{ij}}{|\mathcal{E}|} - \eta\right)^2,
\tag{7}
$$

where $\eta$ is a hyper-parameter that controls the budget for the total number of edges pruned by $t(\cdot)$. Next we show that $\mathcal{L}_e$ will prune edges from spurious subgraph, while preserving the invariant subgraph when sampled from $t(G)$.

**Proposition 2.** *Under Assumption 2, the size constraint loss $\mathcal{L}_e$, when acting as a regularizer for $\mathcal{L}_{GT}$, will prune edges from the spurious subgraph $G_s$, while preserving the invariant subgraph $G_c$.*

Prop. 2 demonstrates that by enforcing graph size constraint, $\mathcal{L}_e$ will only prune spurious edges, thus making the size of $G_s$, i.e., $|G_s|$, to be smaller. Next we show that $\mathcal{L}_e$ provably improves OOD generalization ability by shrinking $|G_s|$.

**Theorem 4.1.** *Let $l((x_i, x_j, y, G); \theta)$ denote the 0-1 loss function for predicting whether edge $e_{ij}$ presents in graph $G$ using $t(\cdot)$, and*

$$L(\theta; D) := \frac{1}{n} \sum_{(x_i, x_j, y, G) \sim D} l((x_i, x_j, y, G); \theta), \forall e_{ij} \in \mathcal{E}.$$

$$L(\theta; S) := \frac{1}{n} \sum_{(x_i, x_j, y, G) \sim S} l((x_i, x_j, y, G); \theta), \forall e_{ij} \in \mathcal{E}. \tag{8}$$

*where $D$ and $S$ represent the training and test set distributions, respectively,c is a constant, and $n$ denotes the sample size. Then, with probability at least $1 - \delta$ and $\forall \theta \in \Theta$, we have:*

$$|L(\theta; D) - L(\theta; S)| \leq 2(c|G_s| + 1)\sqrt{\frac{\ln(4|\Theta|) - \ln(\delta)}{2n}}. \tag{9}$$

We prove Theorem 4.1 in Appendix E.3. Theorem 4.1 establishes an OOD generalization bound that incorporates $|G_s|$ due to domain shifts. When $|G_s| = 0$, Eqn. 9 reduces to the traditional in-distribution generalization bound. From Theorem 4.1, we demonstrate that $\mathcal{L}_e$ enhances the OOD generalization bound by reducing the size of $G_s$ and tightens the generalization bound. Reducing the size of $G_s$ can also help decrease the candidate space of $G_s$, thereby facilitating the subsequent process of *spurious subgraph diversification* to marginalize over all possible spurious patterns and effectively extrapolate to unseen environments, further enhancing the OOD generalization ability.

**Spurious Subgraph Diversification.** Building upon Def. 2, *spurious subgraph diversification* aims to generate diversified spurious patterns and identify the invariant subgraph $G_c$. However, it is challenging to distinguish between $G_c$ and $G_s$ in $t(G)$. Nonetheless, using the results from Prop. 2, it is likely that edges from $G_c$ will exhibit a higher predicted probability using $t(\cdot)$ than edges in $G_s$. Based on this insight, *to accurately identify spurious subgraphs and preserve $G_c$ in the subsequent diversification,* we sort $w_{ij} \in \mathcal{E}$ and consider the lowest $K\%$ of edges as $\widehat{G}_s$ (denoted as $\mathcal{E}_s$). We then align the distribution of these edges as closely as possible to a uniform distribution to diversify the spurious subgraphs. Specifically, we employ the total variation distance to enforce the following regularization:

$$\mathcal{L}_{div} = \mathbb{E}_{\mathcal{G}} \frac{1}{|\mathcal{E}_s|} \sum_{e_{ij} \in \mathcal{E}_s} \mathbb{TV}(e_{ij}, \mathcal{U}) = \mathbb{E}_{\mathcal{G}} \frac{1}{|\mathcal{E}_s|} \sum_{e_{ij} \in \mathcal{E}_s} \left| \bar{w}_{ij} - \frac{1}{|\mathcal{E}_s|} \right|, \tag{10}$$

where $e_{ij} \sim Bernoulli(p_{ij})$, $\bar{w}_{ij}$ denotes the normalized probability of the logits $w_{ij}$, $\mathbb{TV}(\cdot, \cdot)$ denotes the total variation distance, and $\mathcal{U}$ denotes a uniform distribution w.r.t. the estimated spurious subgraph, where each edge $e_{ij} \in \mathcal{E}_s$ appears uniformly at random. The diversification strength is directly related to $|\mathcal{E}_s|$, e.g., when $|\mathcal{E}_s|$ becomes larger, the diversification effect for each edge becomes smaller, while the total strength remains unchanged. The goal of $\mathcal{L}_{div}$ is to increase uncertainty of sampling $e_{ij} \in \mathcal{E}_s$, thereby enhancing the diversity of the spurious subgraphs, meanwhile, the invariant subgraph $G_c$ in each graph $G$ is preserved, as demonstrated in Prop. 2. The overall objective is formulated in below:

$$\mathcal{L} = \mathcal{L}_{GT} + \lambda_1 \mathcal{L}_e + \lambda_2 \mathcal{L}_{div}, \tag{11}$$

here $\lambda_i, i \in \{1, 2\}$ are hyperparameters that balance the contribution of each component to the overall objective. Next we present the following main theorem to demonstrate that $t^*(G)$ after optimizing the loss object $\mathcal{L}$ in Eqn. 11 will correctly identify $G_c$, thus achieve graph OOD generalization under distribution shifts.

**Theorem 4.2.** *Let $\Theta^* = \arg\inf_\Theta \mathcal{L}(\Theta)$, where $\Theta^* = \{\rho^*(\cdot), h^*(\cdot), t^*(\cdot)\}$. For any graph $G$ with target label $y \in \mathcal{Y}$, we have $G_c \approx \mathbb{E}_G[t^*(G)]$, i.e., optimizing the objective function $\mathcal{L}(\Theta)$ will lead to the optimal learnable data transformation function $t^*(\cdot)$. Consequently, sampling from $t^*(G)$ in expectation will retain only the invariant subgraph $G_c$, which remains stable and sufficiently predictive for the target label $y$.*

The proof of Theorem 4.2 is provided in Appendix E.4. It is important to note that while Prop. 2 highlights the inclusion of the invariant subgraph $G_c$, it does not eliminate the possibility of retaining spurious edges, which may affect the method's effectiveness. However, Theorem 4.2 demonstrates that it is able to retain only $G_c$ by sampling from $t^*(G)$. Intuitively, the diversification process reinforces the edges in $G_c$, as $t^*(G)$ consistently includes $G_c$, while simultaneously weakening the spurious edges due to the spurious edge pruning and diversification process. This approach ensures the identification of invariant subgraph $G_c$, and generalization capability of our proposed method.

## 5 DISCUSSION

Although adopting a learnable data transformation function, or a subgraph selector $t(\cdot)$ is not a novel concept in the literature, our study provides a new perspective on how to utilize $t(\cdot)$ for identifying the invariant subgraph $G_c$ for OOD generalization on graphs. Specifically, in previous works such as Li et al. (2022b); Wu et al. (2022c), the learned subgraph is utilized for environment inference and environment generation respectively to learn an *equipredictive* classifier for OOD generalization; In Miao et al. (2022), $t(\cdot)$ is employed to identify $G_c$ based on the information bottleneck principle (Tishby & Zaslavsky, 2015). Chen et al. (2022) leverages $t(\cdot)$ to identify $G_c$ through supervised contrastive learning, while Sui et al. (2023) uses the learned subgraph for distribution perturbation to enhance model robustness; In Gui et al. (2023), $t(\cdot)$ is utilized to isolate $G_c$ under the regularization conditions $G_c \perp\!\!\!\perp E$ and $G_s \perp\!\!\!\perp Y$, where the environments $E$ are assumed observable. Lu et al. (2024) utlizes subgraph selector to extract graph rationals, followed by generating virtual samples by perturbing these substructures using Extreme Value Theory (Haan & Ferreira, 2006). In Jia et al. (2024), the subgraph selector is utilized to separate invariant and spurious subgraphs, followed by invariant mixup and environmental mixup to augment the training distribution. In contrast, our approach is fundamentally different from all previous works in how to utilize $t(\cdot)$ to identify invariant subgraphs. By leveraging spurious subgraph diversification with graph size constraint, our method is able to marginalizing over diversified (enriched) spurious patterns, effectively extrapolate to unseen environments, and identifies $G_c$ using $t^*(\cdot)$ for OOD generalization.

## 6 RELATED WORK

**Invariant learning.** Recently, there has been growing attention on the graph-level representations under distribution shifts from the perspective of invariant learning. Some works focus on environment inference (Yang et al., 2022; Li et al., 2022b) or augmentation strategies (Wu et al., 2022c; Liu et al., 2022; Zhuang et al., 2023; Sui et al., 2023; Li et al., 2024). Another line of works employs alternative strategies to identify $G_c$ without tackling the hidden environment labels (Chen et al., 2022; 2023a; Miao et al., 2022; Yu et al., 2021; 2020), Most of these works assume a causal data generating process that assumes the existence of an invariant subgraph $G_c$ causally related to the target label $Y$, which remains invariant across different distribution shifts. In this work, we also adopt this assumption, and our approach aims to generate diversified environments without affecting the invariant substructures.

**Graph data augmentation.** Recent studies (Rong et al., 2019; Wang et al., 2021; Han et al., 2022) have introduced various graph data augmentation methods to enhance the performance of semi-supervised node classification tasks. Despite these advancements, these methods mainly focus on improving performance within the training distribution and do not directly target OOD data. Inspired by causality (Peters et al., 2016) and invariant learning principles (Arjovsky et al., 2020; Kreuzer et al., 2021), recent works have shifted focus towards environment augmentation, operating both at the input level (Wu et al., 2022c) and latent level (Liu et al., 2022), to generate interpolated (Wu et al., 2022c; Liu et al., 2022; Zhuang et al., 2023) or extrapolated environments (Sui et al., 2023; Li et al., 2024), which typically rely on masking matrices produced by subgraph selectors. Meanwhile, our study introduces spurious subgraph diversification, which enriches the generated spurious patterns, allowing our method to extrapolate to unseen environments more effectively, and capture the stable relationship $\mathbb{P}(Y \mid G_c)$ more accurately.

## 7 EXPERIMENTS

In this section, we evaluate the effectiveness of iSSD on both synthetic datasets and real-world datasets, and answer the following research questions.

- **(RQ1)** How does our method perform compared with SOTA baselines?
- **(RQ2)** How do the individual components and hyperparameters in iSSD affect the overall performance?
- **(RQ3)** Can the optimal learnable data transformation function $t^*(G)$ correctly identify $G_c$?
- **(RQ4)** Do edges in $G_c$ predicted by $t(\cdot)$ exhibit higher probability scores than edges in $G_s$?
- **(RQ5)** How do different GNN architectures impact the OOD performance?

More experimental results including hyperparameter analysis and visualizations are presented in Appendix H.

### 7.1 EXPERIMENTAL SETUP

**Datasets.** We adopt GOOD datasets (Gui et al., 2022), OGBG-Molbbbp datasets (Hu et al., 2020; Wu et al., 2018), and DrugOOD datasets (Ji et al., 2022) to comprehensively evaluate the OOD generalization performance of our proposed framework. More details on these datasets are provided in Appendix H.

**Baselines.** Besides ERM (Vapnik, 1995), we compare our method against four lines of OOD baselines: (1) OOD algorithms on Euclidean data, including IRM (Arjovsky et al., 2020), VREx (Krueger et al., 2021), and GroupDRO (Sagawa et al., 2019); (2) Diverse feature learning methods, including RSC (Huang et al., 2020), and DivCLS (Teney et al., 2022). (3) graph-specific OOD algorithms, including DIR (Wu et al., 2022c), GSAT (Miao et al., 2022), GREA (Liu et al., 2022), DisC (Fan et al., 2022), CIGA (Chen et al., 2022), and AIA (Sui et al., 2023); and (4) graph data augmentation methods, including DropEdge (Rong et al., 2019), $\mathcal{G}$-Mixup (Han et al., 2022), FLAG (Kong et al., 2022), and LiSA (Yu et al., 2023). Details of the baseline setup are provided in Appendix H.2.

**Evaluation.** We report the ROC-AUC score for GOOD-HIV, OGBG-Molbbbp, and DrugOOD datasets, where the tasks are binary classification. For GOOD-Motif, we use accuracy as the evaluation metric. We run experiments 4 times with different random seeds, select models based on the validation performance, and report the mean and standard deviations on the test set.

Table 1: Performance on synthetic and real-world datasets. Numbers in **bold** indicate the best performance, while the underlined numbers indicate the second best performance.

| Method | GOODMotif | | GOODHIV | | EC50 | | | OGBG-Molbbbp | |
|---|---|---|---|---|---|---|---|---|---|
| | base | size | scaffold | size | scaffold | size | assay | scaffold | size |
| ERM | $68.66_{\pm4.25}$ | $51.74_{\pm2.88}$ | $69.58_{\pm2.51}$ | $59.94_{\pm2.37}$ | $62.77_{\pm2.14}$ | $61.03_{\pm1.88}$ | $64.93_{\pm6.25}$ | $68.10_{\pm1.68}$ | $78.29_{\pm3.76}$ |
| IRM | $70.65_{\pm4.17}$ | $51.41_{\pm3.78}$ | $67.97_{\pm1.84}$ | $59.00_{\pm2.92}$ | $63.96_{\pm3.21}$ | $62.47_{\pm1.15}$ | $72.27_{\pm3.41}$ | $67.22_{\pm1.15}$ | $77.56_{\pm2.48}$ |
| GroupDRO | $68.24_{\pm8.92}$ | $51.95_{\pm5.86}$ | $70.64_{\pm2.57}$ | $58.98_{\pm2.16}$ | $64.13_{\pm1.81}$ | $59.06_{\pm1.50}$ | $70.52_{\pm3.38}$ | $66.47_{\pm2.39}$ | $79.27_{\pm2.43}$ |
| VREx | $71.47_{\pm6.69}$ | $52.67_{\pm5.54}$ | $70.77_{\pm2.84}$ | $58.53_{\pm2.88}$ | $64.23_{\pm1.76}$ | $63.54_{\pm1.03}$ | $68.23_{\pm3.19}$ | $68.74_{\pm1.03}$ | $78.76_{\pm2.37}$ |
| RSC | $46.12_{\pm3.76}$ | $51.70_{\pm5.47}$ | $69.16_{\pm3.23}$ | $61.17_{\pm0.74}$ | $64.82_{\pm2.10}$ | $63.38_{\pm1.16}$ | $74.76_{\pm1.96}$ | $69.01_{\pm2.84}$ | $78.07_{\pm3.89}$ |
| DivCLS | $54.24_{\pm8.22}$ | $41.01_{\pm1.98}$ | $69.17_{\pm3.62}$ | $61.59_{\pm2.23}$ | $64.31_{\pm2.16}$ | $63.89_{\pm1.51}$ | $74.79_{\pm4.64}$ | $68.04_{\pm3.27}$ | $77.62_{\pm1.90}$ |
| DropEdge | $45.08_{\pm4.46}$ | $45.63_{\pm4.61}$ | $70.78_{\pm1.38}$ | $58.53_{\pm1.26}$ | $63.91_{\pm2.56}$ | $61.93_{\pm1.41}$ | $73.79_{\pm4.06}$ | $66.49_{\pm1.55}$ | $78.32_{\pm3.44}$ |
| $\mathcal{G}$-Mixup | $59.66_{\pm7.03}$ | $52.81_{\pm6.73}$ | $70.01_{\pm2.52}$ | $59.34_{\pm2.43}$ | $61.90_{\pm2.08}$ | $61.06_{\pm1.74}$ | $69.28_{\pm1.36}$ | $67.44_{\pm1.62}$ | $78.55_{\pm4.16}$ |
| FLAG | $61.12_{\pm5.39}$ | $51.66_{\pm4.14}$ | $68.45_{\pm2.30}$ | $60.59_{\pm2.95}$ | $64.98_{\pm0.87}$ | $64.28_{\pm0.54}$ | $74.91_{\pm1.18}$ | $67.69_{\pm2.36}$ | $79.26_{\pm2.26}$ |
| LiSA | $54.59_{\pm4.81}$ | $53.46_{\pm3.41}$ | $70.38_{\pm1.45}$ | $52.36_{\pm3.73}$ | $62.60_{\pm3.62}$ | $60.96_{\pm1.07}$ | $69.73_{\pm0.62}$ | $68.11_{\pm0.52}$ | $78.62_{\pm3.74}$ |
| DIR | $62.07_{\pm8.75}$ | $52.27_{\pm4.56}$ | $68.07_{\pm2.29}$ | $58.08_{\pm2.31}$ | $63.91_{\pm2.92}$ | $61.91_{\pm3.92}$ | $66.13_{\pm3.01}$ | $66.86_{\pm2.25}$ | $76.40_{\pm4.43}$ |
| DisC | $51.08_{\pm3.08}$ | $50.39_{\pm1.15}$ | $68.07_{\pm1.75}$ | $58.76_{\pm0.91}$ | $59.10_{\pm5.69}$ | $57.64_{\pm1.57}$ | $61.94_{\pm7.76}$ | $67.12_{\pm2.11}$ | $56.59_{\pm10.09}$ |
| CAL | $65.63_{\pm4.29}$ | $51.18_{\pm5.60}$ | $67.37_{\pm3.61}$ | $57.95_{\pm2.24}$ | $65.03_{\pm1.12}$ | $60.92_{\pm2.02}$ | $74.93_{\pm5.12}$ | $68.06_{\pm2.60}$ | $79.50_{\pm4.81}$ |
| GREA | $56.74_{\pm9.23}$ | $54.13_{\pm10.02}$ | $67.79_{\pm2.56}$ | $60.71_{\pm2.20}$ | $64.67_{\pm1.43}$ | $62.17_{\pm1.78}$ | $71.12_{\pm1.87}$ | $69.72_{\pm1.66}$ | $77.34_{\pm3.52}$ |
| GSAT | $62.80_{\pm11.41}$ | $53.20_{\pm8.35}$ | $68.66_{\pm1.35}$ | $58.06_{\pm1.98}$ | $65.12_{\pm1.07}$ | $61.90_{\pm2.12}$ | $74.77_{\pm4.31}$ | $66.78_{\pm1.45}$ | $75.63_{\pm3.83}$ |
| CIGA | $66.43_{\pm11.31}$ | $49.14_{\pm8.34}$ | $69.40_{\pm2.39}$ | $59.55_{\pm2.56}$ | $\underline{65.42}_{\pm1.53}$ | $\underline{64.47}_{\pm0.73}$ | $74.94_{\pm1.91}$ | $64.92_{\pm2.09}$ | $65.98_{\pm3.31}$ |
| AIA | $\underline{73.64}_{\pm5.15}$ | $\underline{55.85}_{\pm7.98}$ | $71.15_{\pm1.81}$ | $61.64_{\pm3.37}$ | $64.71_{\pm0.50}$ | $63.43_{\pm1.35}$ | $\underline{76.01}_{\pm1.18}$ | $\mathbf{70.79}_{\pm1.53}$ | $81.03_{\pm5.15}$ |
| iSSD | $\mathbf{91.48}_{\pm0.40}$ | $\mathbf{66.53}_{\pm8.55}$ | $\mathbf{71.84}_{\pm0.61}$ | $\mathbf{64.99}_{\pm1.63}$ | $\mathbf{67.56}_{\pm0.34}$ | $\mathbf{65.46}_{\pm0.88}$ | $\mathbf{78.01}_{\pm0.42}$ | $\underline{70.32}_{\pm1.73}$ | $\mathbf{81.59}_{\pm5.35}$ |

### 7.2 EXPERIMENTAL RESULTS

In this section, we report the main results on both synthetic and real-world datasets.

**Synthetic datasets.** The GOOD-Motif datasets fully align with our assumptions, making them a suitable benchmark for evaluating the effectiveness of our proposed framework iSSD. Our approach outperforms second-best method AIA by $24.19\%$ and $19.13\%$ in Motif-Base and Motif-Size datasets respectively. This demonstrates the excellent environment extrapolation capability of iSSD, utilizing spurious subgraph diversification to generate randomized spurious edges. While diverse feature learning methods such as RSC (Huang et al., 2020) and DivCLS (Teney et al., 2022) achieve strong performance on real-world datasets, their performance is suboptimal on the GOOD-Motif datasets. This may be because these methods attempt to learn diverse features, while in GOOD-Motif datasets only one invariant subgraph is causally related to the target, potentially leading them to capture patterns in $G_s$ that are not generalizable. In contrast, our method employs spurious subgraph diversification to generate enriched negative feedbacks, thereby weakening spurious patterns and enabling the model to effectively focus on the causal patterns. Notably, the in-distribution performance of ERM on Motif-Base dataset is $92.60\%$ (Gui et al., 2022), while our approach achieves a comparable result of $91.48\%$, further demonstrating the superiority of iSSD in learning domain-invariant features.

**Real-world datasets.** In real-world datasets, which present more complex and realistic distribution shifts, many graph OOD algorithms exhibit instability, occasionally underperforming ERM. In contrast, by effectively incorporating environment augmentation strategy with graph size constraint, our approach consistently achieves stable and superior performance across a diverse set of distribution shifts, and outperform the second-best method by an average of $2.38\%$ in 7 real-world datasets.

### 7.3 ABLATION STUDY

In this section, we evaluate the impact of $\mathcal{L}_e$ and $\mathcal{L}_{div}$ using the GOODMotif and GOODHIV datasets by setting $\lambda_1 = 0$ or $\lambda_2 = 0$ in Eqn. 11 to observe the impacts on model performance.

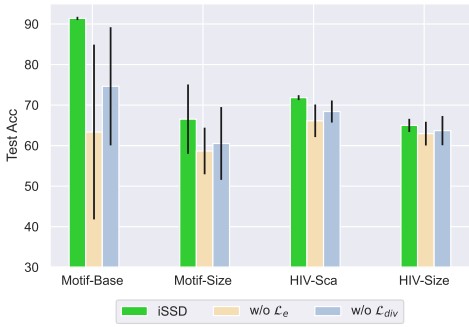

As illustrated in Figure 1, removing either $\mathcal{L}_e$ or $\mathcal{L}_{div}$ leads to a significant drop in test performance across all datasets, and a larger variance. The removal of $\mathcal{L}_e$ results in a more pronounced decline, as this regularization is crucial for $t^*(G)$ to prune spurious edges and reduce the candidate space for the spurious edges to be diversified. However, even with $\mathcal{L}_e$, some spurious edges may still persist within $t^*(G)$, potentially hindering the model's OOD generalization ability. By employing both $\mathcal{L}_e$ and $\mathcal{L}_{div}$, iSSD effectively covers all possible spurious patterns and extrapolate to unseen environments, and achieve superior OOD generalization performance across all four datasets.

Figure 1: Ablation study on $\mathcal{L}_e$ and $\mathcal{L}_{div}$.

### 7.4 HYPER-PARAMETER SENSITIVITY

We study the impact of hyperparameter sensitivity on the edge budget $\eta$ in $\mathcal{L}_e$ and the $K\%$ edges with the lowest probability for $\mathcal{L}_{div}$. Additionally, we investigate the effects of varying the penalty weights for $\mathcal{L}_e$ and $\mathcal{L}_{div}$ (i.e., $\lambda_1$ and $\lambda_2$).

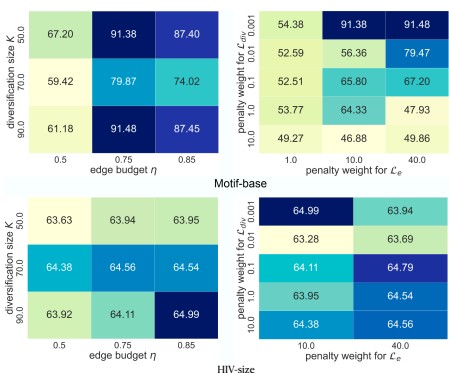

As illustrated in Figure 2, an unsuitable choice of $\eta$ can negatively impact test performance, e.g., in the GOOD-Motif dataset with *base* split, setting $\eta = 0.5$ may prune too many edges, potentially corrupting $G_c$ and consequently reducing test performance. However, with a suitable $\eta$, test performance remains stable across different values of $K$. Notably, a larger $K$ (e.g., $K = 90$) consistently leads to optimal performance, highlighting the effectiveness of spurious subgraph diversification. Readers may raise concerns that a large value of $K$ could also corrupt the invariant substructure $G_c$, seemingly contradicting the optimal test performance observed at $K = 90$. This discrepancy arises because the penalty weight $\lambda_2$ for $\mathcal{L}_{div}$ is smaller than $\lambda_1$, reducing its impact on $G_c$. As shown

Figure 2: Hyperparameter sensitivity.

in Figure 2, when $\lambda_2 \in \{1, 10\}$, test performance declines dramatically for $\forall K \in \{50, 70, 90\}$, supporting our analysis above. Regarding real-world datasets, such as GOODHIV-size and other datasets in Appendix H.3, the test OOD performance demonstrates stability across various hyperparameters, underscoring the robustness of our algorithm.

## 7.5 IN-DEPTH ANALYSIS

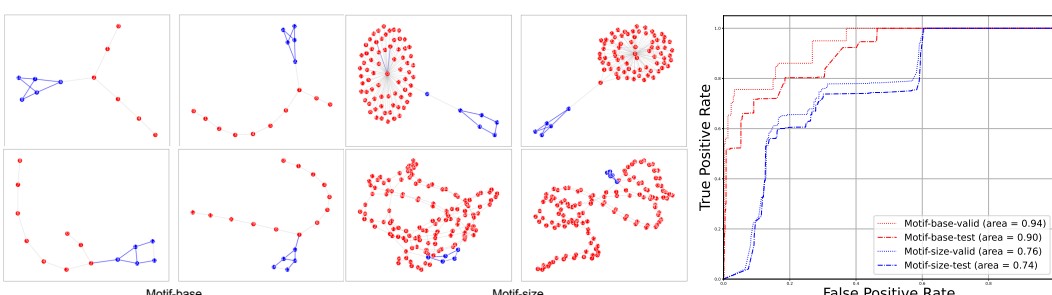

(a). Visualizations on learned subgraph by $t^*(\cdot)$, where blue nodes are ground-truth nodes in $G_c$, and red nodes are ground-truth nodes in $G_s$. The highlighted blue edges are top-K edges predicted by $t^*(\cdot)$, where $K$ is the number of ground-truth causal edges.

(b). The ROC-AUC curve for predicted edges and ground-truth edges on GOODMotif-base and GOODMotif-size datasets.

Figure 3: Empirical visualization and analysis on $t^*(\cdot)$.

**Can $t^*(\cdot)$ identify $G_c$?** To verify whether $t^*(\cdot)$ can indeed identify $G_c$, we conduct experiments using GOOD-Motif datasets with both *base* and *size* splits. These synthetic datasets are suitable for this analysis as they provide ground-truth labels for edges and nodes that are causally related to the targets. First, we collect the predicted probability score and target label for each edge from $t^*(\cdot)$ for correctly predicted samples and plot the ROC-AUC curve for both the validation and test sets across the two datasets. As illustrated in Figure 3(b), the AUC scores for both datasets exhibit high values, demonstrating that $t^*(\cdot)$ accurately identifies $G_c$, which is consistent with the theoretical insights provided in Theorem 4.2. Figure 3(a) illustrates some visualization results using $t^*(\cdot)$, demonstrating that $t^*(\cdot)$ correctly identify causal edges from $G_c$. More visualization results for the identified edges using $t^*(\cdot)$ are provided in Appendix H.3.

**Do edges in $G_c$ exhibit a higher probability than edges in $G_s$?** We assess the probability scores and ranking of edges in $G_c$ compared to those in $G_s$ using the GOOD-Motif datasets. Specifically, we plot the average probability and ranking of edges in $G_c$ over the first 40 epochs (excluding the first 10 epochs for ERM pretraining), using the ground-truth edge labels. As shown in Figure 4, for both the Motif-base and Motif-size datasets, the causal edges in $G_c$ exhibit

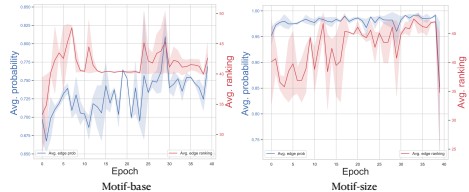

Figure 4: Avg. probability and ranking of edges in $G_c$ for every training epoch.

high probability scores, ranking among the top $50\%$ in both datasets. This confirms the validity of using the lowest $K\%$ probability edges to preserve the invariant subgraphs.

**How do different GNN encoders affect the model performance?** We examine the effect of using different GNN encoders, specifically GCN (Kipf & Welling, 2017) and GIN (Xu et al., 2018), with the same hidden dimensions and number of layers as $h(\cdot)$. As illustrated in Figure 5, across all four datasets, employing GIN as the feature encoder leads to a increase in test performance. This is likely due to GIN's higher expressivity than GCN (Xu et al., 2018), being as powerful as the 1-WL test (Leman & Weisfeiler, 1968), which allows it to generate more distinguishable features compared to GCN. These enhanced features benefits the optimization of $t(\cdot)$, thereby improving the identification of $G_c$ for OOD generalization. This also highlights another advantage of iSSD: utilizing a GNN encoder with enhanced expressivity may further facilitate OOD gen-

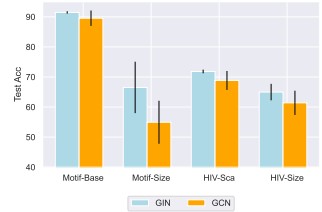

Figure 5: Test performance with different GNN encoders.

eralization by more accurately identifying $G_c$ through $t(\cdot)$, which also provides intrinsic interpretability.

## 8 CONCLUSION

In this work, we focus on the trade-off between generating diverse environments and preserving invariant substructures. To overcome the limitations in existing studies, we proposed *spurious subgraph diversification*, which randomizes the generation of spurious subgraphs to encourage diversity of the generated environments, along with a graph size constraint to reduce the search space of spurious subgraphs for more effective environment extrapolation. Our theoretical analysis and extensive experiments on both synthetic and real-world datasets demonstrate the superiority of our approach for graph OOD generalization.

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

# APPENDIX

## A  NOTATIONS

We present a set of notations used throughout our paper for clarity. Below are the main notations along with their definitions.

Table 2: Notation Table

| Symbols | Definitions |
|---|---|
| $\mathcal{G}$ | Set of graph datasets |
| $\mathcal{E}_{\text{tr}}$ | Set of environments used for training |
| $\mathcal{E}_{\text{all}}$ | Set of all possible environments |
| $G$ | An undirected graph with node set $\mathcal{V}$ and edge set $\mathcal{E}$ |
| $\mathcal{V}$ | Node set of graph $G$ |
| $\mathcal{E}$ | Edge set of graph $G$ |
| $\mathbf{A}$ | Adjacency matrix of graph $G$ |
| $\mathbf{X}$ | Node feature matrix of graph $G$ |
| $D$ | Feature dimension of node features in $\boldsymbol{X}$ |
| $G_c$ | Invariant subgraph of $G$ |
| $G_s$ | Spurious subgraph of $G$ |
| $\widehat{G}_c$ | Estimated invariant subgraph |
| $\widehat{G}_s$ | Estimated spurious subgraph |
| $|G|$ | The number of edges in graph $G$. |
| $E$ | Environmental variable affecting $S$ |
| $Y$ | Target label variable |
| $[K]$ | Index set $\{1, 2, \cdots, K\}$ |
| $\mathbf{w}$ | A vector |
| $\mathbf{W}$ | A matrix |
| $W$ | A random variable |
| $\mathcal{W}$ | A set |
| $f = \rho \circ h$ | A GNN model comprising encoder $h(\cdot)$ and classifier $\rho(\cdot)$ |
| $t(\cdot)$ | Learnable data transformation function for structural modifications |
| $\widetilde{G} \sim t(\cdot)$ | A view sampled from $t(\cdot)$, e.g., $\widetilde{G} \sim t(\cdot)$. We may use $t(G)$ to denote a sampled view from $G$ via $t(\cdot)$, e.g., $I(G; t(G))$ |
| $\mathbf{h}_v$ | Representation of node $v \in \mathcal{V}$ of graph $G$ |

## B  MORE BACKGROUND AND PRELIMINARIES

**Graph Neural Networks.** In this work, we adopt message-passing GNNs for graph classification due to their expressiveness. Given a simple and undirected graph $G = (\mathbf{A}, \mathbf{X})$ with $n$ nodes and $m$ edges, where $\mathbf{A} \in \{0, 1\}^{n \times n}$ is the adjacency matrix, and $\mathbf{X} \in \mathbb{R}^{n \times d}$ is the node feature matrix with $d$ feature dimensions, the graph encoder $h : \mathbb{G} \to \mathbb{R}^h$ aims to learn a meaningful graph-level representation $h_G$, and the classifier $\rho : \mathbb{R}^h \to \mathbb{Y}$ is used to predict the graph label $\widehat{Y}_G = \rho(h_G)$. To obtain the graph representation $h_G$, the representation $\mathbf{h}_v^{(l)}$ of each node $v$ in a graph $G$ is iteratively updated by aggregating information from its neighbors $\mathcal{N}(v)$. For the $l$-th layer, the updated representation is obtained via an AGGREGATE operation followed by an UPDATE operation:

$$\mathbf{m}_v^{(l)} = \text{AGGREGATE}^{(l)} \left( \left\{ \mathbf{h}_u^{(l-1)} : u \in \mathcal{N}(v) \right\} \right), \tag{12}$$

$$\mathbf{h}_v^{(l)} = \text{UPDATE}^{(l)} \left( \mathbf{h}_v^{(l-1)}, \mathbf{m}_v^{(l)} \right), \tag{13}$$

where $\mathbf{h}_v^{(0)} = \mathbf{x}_v$ is the initial node feature of node $v$ in graph $G$. Then GNNs employ a READOUT function to aggregate the final layer node features $\left\{ \mathbf{h}_v^{(L)} : v \in \mathcal{V} \right\}$ into a graph-level representation $\mathbf{h}_G$:

$$\mathbf{h}_G = \text{READOUT} \left( \left\{ \mathbf{h}_v^{(L)} : v \in \mathcal{V} \right\} \right). \tag{14}$$

## C ADDITIONAL RELATED WORK

**OOD Generalization.** OOD generalization is a critical challenge in machine learning, where models trained on a specific data distribution often fail to generalize well to unseen distributions. Several approaches have been proposed to address this issue, including domain generalization, distributional robustness optimization (DRO), and invariance learning. *Domain generalization* aims to learn features that are invariant across different domains or environments. Previous studies, such as Ganin et al. (2016); Sun & Saenko (2016); Li et al. (2018); Dou et al. (2019), regularize the learned features to be domain-invariant. *DRO* methods focus on training models to perform robust against the worst-case scenarios among diverse data groups. Namkoong & Duchi (2016); Hu et al. (2018); Sagawa et al. (2019) regularize models to be robust to mild distributional perturbations of the training distributions, expecting the models to perform well in unseen test environments. Building upon this, Liu et al. (2022) Zhang et al. (2022b) and Yao et al. (2022) propose advanced strategies to improve robustness by assuming that models trained with ERM have a strong reliance on spurious features. *Invariance learning* leverages the theory of causality Peters et al. (2016); Pearl (2009) and introduces causal invariance to the learned representations. The Independent Causal Mechanism (ICM) assumption in causality states that the conditional distribution of each variable given its causes does not inform or influence other conditional distributions. Despite changes to the intervened variables, the conditional distribution of intervened variables and the target variable remains invariant. Arjovsky et al. (2020) proposes the framework of Invariant Risk Minimization (IRM) that allows the adoption of causal invariance in deep neural networks, inspiring various invariant learning works such as Parascandolo et al. (2020); Mahajan et al. (2021); Wald et al. (2021); Ahuja et al. (2020; 2021). These works aim to discard spurious signals while keeping causally invariant signals. However, most of these methods require explicit environment partitions within the dataset, which is often impractical in real-world scenarios. To address this limitation, EIIL (Creager et al., 2021) and Heterogeneous Risk Minimization (HRM) (Liu et al., 2021) propose methods for invariance learning without explicit environment partitions.

**OOD Generalization on Graphs.** Recently, there has been a growing interest in learning graph-level representations that are robust under distribution shifts, particularly from the perspective of invariant learning. MoleOOD (Yang et al., 2022) and GIL (Li et al., 2022b) propose to infer environmental labels to assist in identifying invariant substructures within graphs. DIR (Wu et al., 2022c), GREA (Liu et al., 2022) and iMoLD (Zhuang et al., 2023) employ environment augmentation techniques to facilitate the learning of invariant graph-level representations. These methods typically rely on the explicit manipulation of unobserved environmental variables to achieve generalization across unseen distributions. AIA (Sui et al., 2023) employs an adversarial augmenter to explore OOD data by generating new environments while maintaining stable feature consistency. To circumvent the need for environmental inference or augmentation, CIGA (Chen et al., 2022) and GALA (Chen et al., 2023a) utilizes supervised contrastive learning to identify invariant subgraphs based on the assumption that samples sharing the same label exhibit similar invariant subgraphs. LECI (Gui et al., 2023) and G-Splice (Li et al., 2023) assume the availability of environment labels, and study environment exploitation strategies for graph OOD generalization. LECI (Gui et al., 2023) proposes to learn a causal subgraph selector by jointly optimizing label and environment causal independence, and G-Splice (Li et al., 2023) studies graph and feature space extrapolation for environment augmentation, which maintains causal validity. On the other hand, some works do not utilize the invariance principle for graph OOD generalization. DisC (Fan et al., 2022) initially learns a biased graph representation and subsequently focuses on unbiased graphs to discover invariant subgraphs. GSAT (Miao et al., 2022) utilizes information bottleneck principle (Tishby & Zaslavsky, 2015) to learn a minimal sufficient subgraph for GNN explainability, which is shown to be generalizable under distribution shifts. OOD-GNN (Li et al., 2022a) proposes to learn disentangled graph representation by computing global weights of all data. iSSD also holds significant potential for node-level and link-level OOD generalization. For example, iSSD could be extended by diversifying the $K$-hop subgraphs for each

node to mitigate spurious correlations. Given the rich node features typically available in node-level and link-level tasks, further diversification of node features could serve as an enhanced augmentation strategy. Similarly, constraining the size of ego-networks and adjusting the information density of node features could improve generalization performance in these tasks, which presents promising future directions.

**Diverse feature learning.** Recent studies have shown that ERM tends to encourage models to learn the simplest predictive features (Hermann & Lampinen, 2020; Kalimeris et al., 2019; Neyshabur et al., 2014; Pezeshki et al., 2021). This simplicity bias causes the models to rely on simple (spurious) but non-causal features, ignoring more complex patterns that might be equally predictive. To address this challenge, RSC (Huang et al., 2020) employs a self-challenging mechanism to force the model to learn diverse patterns by discarding dominant features, while DivCLS (Teney et al., 2022) constructs diverse features by training a collection of classifiers with diversity regularization. Additionally, Zhang et al. (2022a); Chen et al. (2023b) adopt DRO to iteratively explore new features. These works primarily focus on Euclidean data and take a model-centric approach, whereas our proposed method is a data-centric approach specifically designed for graph data, which generates diverse negative feedbacks (spurious subgraphs), from which causal features can be identified.

## D    ALGORITHMIC PSEUDOCODE

In this section, we provide the pseudocode of our proposed framework `iSSD`. Our codes will be made publicly available.

---

**Algorithm 1** The proposed method

---

**Input:** Graph dataset $\mathcal{G}$, epochs $E$, learning rates $\eta$, hyperparameters $\lambda_1, \lambda_2$
**Output:** Optimized GNN model $f^* = \rho^* \circ h^*$, and the learnable data transformation function $t^*(\cdot)$.
**Initialize:** GNN encoder $h(\cdot)$, classifier $\rho(\cdot)$, and the learnable data transformation $t(\cdot)$.
**for** epoch $e = 1$ **to** $E$ **do**
    **for** each minibatch $\mathcal{B} \in \mathcal{G}$ **do**
        Calculate $w_{ij}$ using Eqn. 4 for each graph $G \in \mathcal{B}$
        Calculate $\mathcal{L}_e$ using Eqn. 7
        Calculate $\mathcal{L}_{div}$ using Eqn. 10
        Sample $t(G)$ using the learnable data transformation $t(\cdot)$ for each $G \in \mathcal{B}$
        Calculate cross-entropy loss $\mathcal{L}_{GT}$ using Eqn. 6 with $t(G)$
        Compute the total loss $\mathcal{L} = \mathcal{L}_{GT} + \lambda_1 \mathcal{L}_e + \lambda_2 \mathcal{L}_{div}$
        Perform backpropagation to update the parameters of $h(\cdot)$, $\rho(\cdot)$, and $t(\cdot)$
    **end for**
**end for**

---

## E    PROOFS OF THEORETICAL RESULTS

### E.1    PROOF OF THEOREM 3.1

*Proof.* We begin by decomposing the mutual information $I(G;Y)$ as $\lambda I(G;Y) + (1-\lambda)I(G;Y)$. For the first term, denote $\widetilde{G} \sim t(G)$, the following inequalities hold:

$$I(G;Y) \geq I(\widetilde{G};Y) \geq I(f(\widetilde{G});Y) = H(Y) - H(Y \mid f(\widetilde{G})). \tag{15}$$

The first and second inequalities leverage the data processing inequality (Cover & Thomas, 2006). Considering the Markov chain $G \to \widetilde{G} \to f(\widetilde{G}) \to Y$, the mutual information $I(\widetilde{G}, Y)$ cannot exceed $I(G;Y)$. Similarly, the transformation $f(\cdot)$, which is constrained by the 1 Weisfeiler-Lehman test, might not distinguish certain isomorphic substructures. This results in $I(f(\widetilde{G});Y)$ potentially being lower than $I(\widetilde{G};Y)$ due to reduced distinguishability of these substructures.

For the second term, we can derive the following inequality:

$$I(G;Y) \geq I(\widetilde{G};Y) = H(\widetilde{G}) - H(\widetilde{G} \mid Y). \tag{16}$$

By combining Eqn. 15 and Eqn. 16, we get:

$$I(G;Y) \geq -\lambda H(Y \mid f(\widetilde{G})) + (1-\lambda)H(\widetilde{G}) - (1-\lambda)H(\widetilde{G} \mid Y). \tag{17}$$

We thus conclude the proof for Theorem 3.1. $\qquad\square$

### E.2 PROOF OF PROPOSITION 2

*Proof.* We begin by expanding the cross-entropy loss $\mathcal{L}_{GT}$ as:

$$\mathcal{L}_{GT} = -\mathbb{E}_{\mathcal{G}}\left[\log \mathbb{P}(Y \mid f(\widetilde{G}))\right], \tag{18}$$

where $\widetilde{G} \sim t(G)$. Supposing that $|\widetilde{G}| > |G_c|$, which can be controlled by the hyperparameter $\eta$ in Eqn. 7, further assume that $\widetilde{G}$ does not include the invariant subgraph $G_c$. Let a subgraph $g$ be substracted from $\widetilde{G}$ and $|g| = |G_c|$, we then define a new subgraph $G' = \widetilde{G} \setminus g$, and we add $G_c$ to $G'$ to form the new graph $G' \cup G_c$.

Under Assumption 1, we know that the invariant subgraph $G_c$ holds sufficient predictive power to $Y$, and $G_c$ is more informative to $Y$ than $G_s$ (Assumption 2), therefore including $G_c$ will always make the prediction more certain, i.e.,

$$\mathbb{P}(Y \mid f(G' \cup G_c)) > \mathbb{P}(Y \mid f(G' \cup g)), \forall g \subseteq \widetilde{G}, \tag{19}$$

As a result, $\mathcal{L}_{GT}$ will become smaller. Therefore, we conclude that under the graph size regularization imposed by $\mathcal{L}_e$, the optimal solution $\widetilde{G} \sim t(G)$ will always include the invariant subgraph $G_c$, while pruning edges from the spurious subgraph $G_s$.

$\qquad\square$

### E.3 PROOF OF THEOREM 4.1

*Proof.* We first formally define the notations in our proof. Let $l((x_i, x_j, y, G); \theta)$ denotes the 0-1 loss for the edge $e_{ij}$ being presented in graph $G$, and

$$
\begin{aligned}
L(\theta; D) &:= \frac{1}{n} \sum_{(x_i, x_j, y, G) \sim D} l\left((x_i, x_j, y, G)\,; \theta\right), \\
L(\theta; S) &:= \frac{1}{n} \sum_{(x_i, x_j, y, G) \sim S} l\left((x_i, x_j, y, G)\,; \theta\right),
\end{aligned}
\tag{20}
$$

where $D$ and $S$ are training and test distribution, $n$ represents the sample size. Furthermore,

$$
\begin{aligned}
L_c(\theta; D) &= \frac{1}{n} \sum_{(x_i, x_j, y, G) \sim D} l((x_i, x_j, y, G_c)\,; \theta), \forall e_{ij} \in G_c. \\
L_s(\theta; D) &= \frac{1}{n} \sum_{(x_i, x_j, y, G) \sim D} l\left((x_i, x_j, y, G_s)\,; \theta\right), \forall e_{ij} \in G_s.
\end{aligned}
\tag{21}
$$

and $L_c(\theta; D)$, $L_c(\theta; S)$ can be similarly defined. $L_c(\theta; S)$ and $L_c(\theta; D)$ will be identically distributed given Assumption 1 that $G_c$ is stable across different environments, while $L_s(\theta; D)$, $L_s(\theta; S)$ will be distributed differently due to $G_s$. We also assume that: $\mathbb{E}_D[l((x_i, x_j, y, G_c)\,; \theta)] = \mathbb{E}_S[l((x_i, x_j, y, G_c)\,; \theta)]$, as $G_c$ is stable in training and test environments. Finally, we assume:

$$L_s(\theta; D) := c\,|G_s|\,L_c(\theta; D), \tag{22}$$

which implies that $L_s(\theta; \cdot)$ will be proportional to the size of $G_s$, when $|G_s| = 0$ the loss term $L_s(\theta; D) = 0$ ,and the loss mainly arises from the in-distribution loss $L_c(\theta; D)$. Similarly we can define for $L_s(\theta; S)$.

$$
\begin{align}
|L(\theta; D) - L(\theta; S)| &= |L(\theta; D) - L(\theta; S)| \tag{23} \\
&= |L_c(\theta; D) + L_s(\theta; D) - L_c(\theta; S) - L_s(\theta; S)| \tag{24} \\
&= |L_c(\theta; D) - L_c(\theta; S) + L_s(\theta; D) - L_s(\theta; S)| \tag{25} \\
&\leq |L_c(\theta; D) - L_c(\theta; S)| + |L_s(\theta; D) - L_s(\theta; S)| \tag{26} \\
&= |L_c(\theta; D) - L_c(\theta; S)| + c\,|G_s|\,|L_c(\theta; D) - L_c(\theta; S)|\,. \tag{27} \\
&= (c\,|G_s| + 1)\,|L_c(\theta; D) - L_c(\theta; S)|\,. \tag{28}
\end{align}
$$

As in Eqn. 28, $L_c(\theta; \cdot)$ follows the same distribution regardless of the data distributions $D$ or $S$ due to the stability of $G_c$ across different domains, and for any $\theta \in \Theta$, $l((x_i, x_j, y, G_c)$ is bounded in the range $[0, 1]$, we have:

$$
\begin{align}
|L_c(\theta; D) - L_c(\theta; S)| &= |L_c(\theta; D) - \mathbb{E}\left[L_c(\theta; D)\right] + \mathbb{E}\left[L_c(\theta; S)\right] - L_c(\theta; S)| \tag{29} \\
&\leq |L_c(\theta; D) - \mathbb{E}\left[L_c(\theta; D)\right]| + |\mathbb{E}\left[L_c(\theta; S)\right] - L_c(\theta; S)| \tag{30} \\
&\leq |L_c(\theta; D) - \mathbb{E}\left[L_c(\theta; D)\right]| + |\mathbb{E}\left[L_c(\theta; S)\right] - L_c(\theta; S)|\,. \tag{31}
\end{align}
$$

For each term, we can apply Hoeffding's Inequality:

$$\mathbb{P}\left(|\mathbb{E}\left[L_c(\theta; D)\right] - L_c(\theta; D)| \geq \epsilon\right) \leq 2\exp\left(-2\epsilon^2 n\right), \tag{32}$$

$$\mathbb{P}\left(|\mathbb{E}\left[L_c(\theta; D)\right] - L_c(\theta; D)| \geq \epsilon \text{ for any } \theta \in \Theta\right) \leq \sum_{\theta \in \Theta} 2\exp\left(-2\epsilon^2 n\right), \tag{33}$$

thus for the first term $|\mathbb{E}\left[L_c(\theta; D)\right] - L_c(\theta; D)|$, we have

$$\mathbb{P}\left(\exists \theta \in \Theta \text{ such that } |\mathbb{E}\left[L_c(\theta; D)\right] - L_c(\theta; D)| \geq \epsilon\right) \leq 2|\Theta|\exp\left(-2\epsilon^2 n\right). \tag{34}$$

Similarly, for the second term $|\mathbb{E}\left[L_c(\theta; S)\right] - L_c(\theta; S)|$, we have

$$\mathbb{P}\left(\exists \theta \in \Theta \text{ such that } |\mathbb{E}\left[L_c(\theta; S)\right] - L_c(\theta; S)| \geq \epsilon\right) \leq 2|\Theta|\exp\left(-2\epsilon^2 n\right). \tag{35}$$

To upper bound Eqn. 31 with probability $1 - \delta$, for both terms $|\mathbb{E}\left[L_c(\theta; D)\right] - L_c(\theta; D)|$ and $|\mathbb{E}\left[L_c(\theta; S)\right] - L_c(\theta; S)|$, we set the right hand side of Eqn. 34 and Eqn. 35 as $\delta/2$, that is,

$$2|\Theta|\exp\left(-2\epsilon^2 n\right) = \frac{\delta}{2} \Rightarrow \delta = 4|\Theta|\exp\left(-2\epsilon^2 n\right). \tag{36}$$

Therefore, we conclude that with probability at least $1 - \delta$, we have

$$|L_c(\theta; D) - L_c(\theta; S)| \leq 2\sqrt{\frac{\ln(4|\Theta|) - \ln(\delta)}{2n}}. \tag{37}$$

Finally, we get:

$$|L(\theta; D) - L(\theta; S)| \leq 2(c\,|G_s| + 1)\sqrt{\frac{\ln(4|\Theta|) - \ln(\delta)}{2n}}. \tag{38}$$

$\square$

### E.4 PROOF OF THEOREM 4.2

*Proof.* Our proof consists of the following steps.

**Step 1.** We start by decomposing $\mathbb{E}[t^*(G)]$ into two components: the invariant subgraph $G_c$ and a partially retained spurious subgraph $G_s^{\mathcal{P}}$.

$$
\begin{aligned}
\mathbb{E}[t^*(G)] &= \mathbb{E}\left[G_c + G_s^{\mathcal{P}}\right] \\
&= \mathbb{E}\left[G_c\right] + \mathbb{E}\left[G_s^{\mathcal{P}}\right] \\
&= G_c + \mathbb{E}\left[G_s^{\mathcal{P}}\right]
\end{aligned}
\tag{39}
$$

In Eqn. 39, $\mathbb{E}\left[G_c\right] = G_c$ is due to that for any given label $y$, $G_c$ is a constant according to Assumption 1, while $G_s^{\mathcal{P}}$ is a random variable.

**Step 2.** We then model $G_s^{\mathcal{P}}$ as a set of independent edges, and calculate the expected total edge weights of $G_c$ and $G_s^{\mathcal{P}}$ respectively. First, we define $W_c$ as the sum of binary random variables corresponding to the edges in $G_c$. Each edge $e_{ij}$ in $G_c$ is associated with a Bernoulli random variable $X_{ij}$ such that:

$$
W_c = \sum_{e_{ij} \in G_c} X_{ij}.
\tag{40}
$$

Similarly, we define $W_s^{\mathcal{P}}$ as the sum of binary random variables corresponding to the edges in $G_s^{\mathcal{P}}$. Each edge $e_{ij}$ in $G_s^{\mathcal{P}}$ is associated with a Bernoulli random variable $X'_{ij}$ such that:

$$
W_s^{\mathcal{P}} = \sum_{e_{ij} \in G_s^{\mathcal{P}}} X'_{ij}.
\tag{41}
$$

$W_c$ and $W_s^{\mathcal{P}}$ are denoted as random r.v. for the total edge weights of $G_c$ and $G_s^{\mathcal{P}}$.

**Step 3.** We then calculate the expected edge weights $\mathbb{E}[W_c]$ and $\mathbb{E}[W_s^{\mathcal{P}}]$ as following.

$$
\mathbb{E}[W_c] = \mathbb{E}[\sum_{e_{ij} \in G_c} X_{ij}] = \sum_{e_{ij} \in G_c} \mathbb{E}[X_{ij}] = |G_c|,
\tag{42}
$$

$$
\mathbb{E}[W_s^{\mathcal{P}}] = \mathbb{E}[\sum_{e_{ij} \in G_s^{\mathcal{P}}} X'_{ij}] = \sum_{e_{ij} \in G_s^{\mathcal{P}}} \mathbb{E}[X'_{ij}] = \frac{|G_s^{\mathcal{P}}|}{|\mathcal{E}_s|} = \frac{\eta|\mathcal{E}| - |G_c|}{|\mathcal{E}_s|}.
\tag{43}
$$

Here $\mathcal{E}_s$ is the set of diversified edges, i.e., the bottom $K\%$ edges predicted by $t(\cdot)$, $\eta|\mathcal{E}|$ is the total edge number limits due to $\mathcal{L}_e$. In Eqn. 42, $\mathbb{E}[X_{ij}] = 1, \forall e_{ij} \in G_c$ is due to that $\mathbb{P}(X_{ij}) = 1$, as $t^*(G)$ always include $G_c$ using the results from Prop. 2; In Eqn. 43, $\mathbb{E}[X'_{ij}] = \frac{1}{|\mathcal{E}_s|}, \forall e_{ij} \in G_s^{\mathcal{P}}$, due to that $\mathbb{P}(X'_{ij}) = \frac{1}{|\mathcal{E}_s|}$ enforced by the diversification regularization $\mathcal{L}_{div}$. Therefore, given a suitable $\eta$ that prunes spurious edges from $G_s$, $|\mathcal{E}_s||G_c| \gg \eta|\mathcal{E}| - |G_c|$, i.e., $\mathbb{E}[t^*(G)]$ will be dominated by $G_c$ in terms of edge probability mass, therefore, we conclude that $G_c \approx \mathbb{E}[t^*(G)]$.

$\square$

## F COMPLEXITY ANALYSIS

**Time Complexity.** The time complexity is $\mathcal{O}(CkmF)$, where $k$ is the number of GNN layers, $m$ is the total number of edges in graph $G$, and $F$ is the feature dimensions. Compared to ERM, `iSSD` incurs an additional constant $C > 1$, as it uses a GNN model $t(\cdot)$ for edge selection, and another GNN encoder $h(\cdot)$ for learning feature representations. However, $C$ is a small constant, hence the time cost is on par with standard ERM.

**Space Complexity.** The space complexity for iSSD is $\mathcal{O}(C'|\mathcal{B}|mkF)$, where $|\mathcal{B}|$ denotes the batch size. The constant $C' > 1$ is due to the additional data transformation $t(\cdot)$. As $C'$ is also a small integer, the space complexity of iSSD is also on par with standard ERM.

# G   MORE DISCUSSIONS ON GRAPH COMPOSITION STRATEGIES

Existing methods (Li et al., 2024; Sui et al., 2023; Wu et al., 2022c; Liu et al., 2022) typically adopt an *additive formulation* to compose the estimated invariant and spurious subgraphs, employing a model-based (e.g., GNN) subgraph selector. For example, DIR (Wu et al., 2022c) uses the following formulation for graph composition:

$$\mathcal{E}_{\tilde{c}} = \text{Top}_r(\mathbf{M} \odot \mathbf{A}), \quad \mathcal{E}_{\tilde{s}} = \text{Top}_{1-r}((1 - \mathbf{M}) \odot \mathbf{A}), \tag{44}$$

where $\mathbf{A}$ denotes the adjacency matrix, $\mathbf{M}$ denotes the learnable masking matrix, $\mathcal{E}_{\tilde{c}}$ and $\mathcal{E}_{\tilde{s}}$ represent the edge sets of $\tilde{c}$ and $\tilde{s}$, respectively. $\text{Top}_r(\cdot)$ selects the top $K$ edges, where $K = r \times |\mathcal{E}|$, and $r$ is a hyper-parameter. Similarly, GREA (Liu et al., 2022) adopts an additive approach using node masking matrices to distinguish between invariant and spurious nodes.

$$\mathbf{h}^{(r)} = \mathbf{1}_N^\top \cdot (\mathbf{m} \times \mathbf{H}), \quad \mathbf{h}^{(e)} = \mathbf{1}_N^\top \cdot ((\mathbf{1}_N - \mathbf{m}) \times \mathbf{H}), \tag{45}$$

where $\mathbf{m}$ is the node masking matrix learned by a GNN encoder, $\mathbf{H}$ is the node representations derived from another GNN encoder, $\mathbf{1}_N$ is an $N$-dimensional column vector with all entries equal to 1, and $\mathbf{h}^{(r)}$ and $\mathbf{h}^{(e)} \in \mathbb{R}^d$ are the representation vectors of the rationale subgraph $g^{(r)}$ and the environment subgraph $g^{(e)}$ respectively. AIA (Sui et al., 2023) also adopts a similar form for graph augmentation, where the augmented graph is formulated as:

$$\widetilde{\mathbf{M}} = (\mathbf{1}^a - \mathbf{M}_{\text{sta}}^a) \odot \mathbf{M}_{\text{adv}}^a + \mathbf{M}_{\text{sta}}^a, \tag{46}$$

where $\mathbf{M}_{\text{sta}}$ represents the masking matrix for stable patterns, and $\mathbf{M}_{\text{adv}}$ represents masking matrix for adversarial perturbations for spurious patterns.

All the above methods utilize closed-form graph composition strategies with learnable masking matrices. This can limit the diversity of the generated environments, due to model bias and restricted composition form. To address these challenges, we propose *spurious subgraph diversification* to maximally randomize the generation of spurious subgraphs to encourage diversity of the generated environments, which eliminates the use of masking matrices and closed-form composition strategies.

# H   MORE DETAILS ABOUT EXPERIMENTS

## H.1   DATASETS DETAILS

In our experimental setup, we utilize four datasets: GOOD-HIV, GOOD-Motif, OGBG-MolBBBP, and DrugOOD. The statistics of the datasets are illustrated in Table 3.

**GOOD-HIV** is a molecular dataset derived from the MoleculeNet (Wu et al., 2018) benchmark, where the primary task is to predict the ability of molecules to inhibit HIV replication. The molecular structures are represented as graphs, with nodes as atoms and edges as chemical bonds. Following Gui et al. (2022), We adopt the covariate shift split, which refers to changes in the input distribution between training and testing datasets while maintaining the same conditional distribution of labels given inputs. This setup ensures that the model must generalize to unseen molecular structures that differ in these domain features from those seen during training. We focus on the Bemis-Murcko scaffold (Bemis & Murcko, 1996) and the number of nodes in the molecular graph as two domain features to evaluate our method.

**GOOD-Motif** is a synthetic dataset designed to test structure shifts. Each graph in this dataset is created by combining a base graph and a motif, with the motif solely determining the label. The base graph type and the size are selected as domain features to introduce covariate shifts. By generating

different base graphs such as wheels, trees, or ladders, the dataset challenges the model's ability to generalize to new graph structures not seen during training. We employ the covariate shift split, where these domain features vary between training and testing datasets, reflecting real-world scenarios where underlying graph structures may change.

**OGBG-Molbbbp** is a real-world molecular dataset included in the Open Graph Benchmark (Hu et al., 2020). This dataset focuses on predicting the blood-brain barrier penetration of molecules, a critical property in drug discovery. The molecular graphs are detailed, with nodes representing atoms and edges representing bonds. Following Sui et al. (2023), we create scaffold shift and graph size shift to evaluate our method. Similarly to Gui et al. (2022), the Bemis-Murcko scaffold (Bemis & Murcko, 1996) and the number of nodes in the molecular graph are used as domain features to create scaffold shift and size shift respectively.

**DrugOOD** (Ji et al., 2022) is designed for OOD challenges in AI-aided drug discovery. This benchmark offers three environment-splitting strategies: Assay, Scaffold, and Size. In our study, we adopt the EC50 measurement. Consequently, this setup results in three distinct datasets, each focusing on a binary classification task for predicting drug-target binding affinity.

Table 3: Details about the datasets used in our experiments.

| DATASETS | Split | # TRAINING | # VALIDATION | # TESTING | # CLASSES | METRICS |
|---|---|---|---|---|---|---|
| GOOD-HIV | Scaffold | 24682 | 4113 | 4108 | 2 | ROC-AUC |
| | Size | 26169 | 4112 | 3961 | 2 | ROC-AUC |
| GOOD-Motif | Base | 18000 | 3000 | 3000 | 3 | ACC |
| | Size | 18000 | 3000 | 3000 | 3 | ACC |
| OGBG-Molbbbp | Scaffold | 1631 | 204 | 204 | 2 | ROC-AUC |
| | Size | 1633 | 203 | 203 | 2 | ROC-AUC |
| EC50 | Assay | 4978 | 2761 | 2725 | 2 | ROC-AUC |
| | Scaffold | 2743 | 2723 | 2762 | 2 | ROC-AUC |
| | Size | 5189 | 2495 | 2505 | 2 | ROC-AUC |

## H.2 DETAILED EXPERIMENT SETTING

**GNN Encoder.** For GOOD-Motif datasets, we utilize a 4-layer GIN (Xu et al., 2018) without Virtual Nodes (Gilmer et al., 2017), with a hidden dimension of 300; For GOOD-HIV datasets, we employ a 4-layer GIN without Virtual Nodes, and with a hidden dimension of 128; For the OGBG-Molbbbp dataset, we adopt a 4-layer GIN with Virtual Nodes, and the dimensions of hidden layers is 64; For the DrugOOD datasets, we use a 4-layer GIN without Virtual Nodes. All GNN backbones adopt sum pooling for graph readout.

**Training and Validation.** By default, we use Adam optimizer (Kingma & Ba, 2014) with a learning rate of $1e - 3$ and a batch size of 64 for all experiments. For DrugOOD, GOOD-Motif and GOOD-HIV datasets, our method is pretrained for 10 epochs with ERM, and for other datasets, we do not use ERM pretraining. We employ an early stopping of 10 epochs according to the validation performance for DrugOOD datasets and GOOD-Motif datasets, and do not employ early stopping for other datasets. Test accuracy or ROC-AUC is obtained according to the best validation performance for all experiments. All experiments are run with 4 different random seeds, the mean and standard deviation are reported using the 4 runs of experiments.

**Baseline setup and hyperparameters.** In our experiments, for the GOOD and OGBG-Molbbbp datasets, the results of ERM, IRM, GroupDRO, and VREx are reported from Gui et al. (2022), while the results for DropEdge, DIR, GSAT, CIGA, GREA, FLAG, $\mathcal{G}$-Mixup and AIA on GOOD and OGBG datasets are reported from Sui et al. (2023). To ensure fairness, we adopt the same GIN backbone architecture as reported in Sui et al. (2023). For the EC50 datasets and the diverse feature learning methods (RSC (Huang et al., 2020) and DivCLS (Teney et al., 2022)), we conduct experiments using the provided source codes from the baseline methods. The hyperparameter search is detailed as follows.

For IRM and VREx, the weight of the penalty loss is searched over $\{1e - 1, 1, 1e1, 1e2\}$. For GroupDRO, the step size is searched over $\{1.0, 1e - 1, 1e - 2\}$. The causal subgraph ratio for DIR is searched across $\{1e - 2, 1e - 1, 0.2, 0.4, 0.6\}$. For RSC, the masking ratio is searched over

Table 4: Experimental results on SPMotif datasets with 2 invariant subgraphs in each graph.

| Method | SPMotif ($\#G_c = 2$) | | |
|---|---|---|---|
| | $b = 0.40$ | $b = 0.60$ | $b = 0.90$ |
| ERM | $53.48_{\pm 3.31}$ | $52.59_{\pm 4.61}$ | $56.76_{\pm 8.06}$ |
| IRM | $52.47_{\pm 3.63}$ | $55.62_{\pm 7.90}$ | $48.66_{\pm 2.33}$ |
| VRex | $49.68_{\pm 8.66}$ | $48.89_{\pm 4.79}$ | $47.97_{\pm 2.61}$ |
| GSAT | $59.34_{\pm 7.96}$ | $58.43_{\pm 10.64}$ | $55.68_{\pm 3.18}$ |
| GREA | $64.87_{\pm 5.76}$ | $67.66_{\pm 6.29}$ | $59.40_{\pm 10.26}$ |
| CIGA | $69.74_{\pm 6.81}$ | $71.19_{\pm 2.46}$ | $65.83_{\pm 10.41}$ |
| AIA | $\mathbf{71.61}_{\pm \mathbf{2.09}}$ | $72.01_{\pm 2.13}$ | $58.14_{\pm 4.21}$ |
| iSSD | $70.41_{\pm 7.53}$ | $\mathbf{74.61}_{\pm \mathbf{3.17}}$ | $\mathbf{66.75}_{\pm \mathbf{4.33}}$ |

$\{0.2, 0.3, 0.4\}$. For DivCLS, the number of classifciation headers is searhched over $\{5, 10, 20\}$, and the penalty weight of the diversification loss is searched over $\{1e-1, 1e-2, 1e-3\}$. For DropEdge, the edge masking ratio is seached over: $\{0.1, 0.2, 0.3\}$. For GREA, the weight of the penalty loss is tuned over $\{1e-2, 1e-1, 1.0\}$, and the causal subgraph size ratio is tuned over $\{0.05, 0.1, 0.2, 0.3, 0.5\}$. For GSAT, the causal graph size ratio is searched over $\{0.3, 0.5, 0.7\}$. For CIGA, the contrastive loss and hinge loss weights are searched over $\{0.5, 1.0, 2.0, 4.0, 8.0\}$. For DisC, we search over $q$ in the GCE loss: $\{0.5, 0.7, 0.9\}$. For LiSA, the loss penalty weights are searched over:$\{1, 1e-1, 1e-2, 1e-3\}$. For $\mathcal{G}$-Mixup, the augmented ratio is tuned over $\{0.15, 0.25, 0.5\}$. For FLAG, the ascending steps are set to 3 as recommended in the paper, and the step size is searched over $\{1e-3, 1e-2, 1e-1\}$. For AIA, the stable feature ratio is searched over $\{0.1, 0.3, 0.5, 0.7, 0.9\}$, and the adversarial penalty weight is searched over $\{0.01, 0.1, 0.2, 0.5, 1.0, 3.0, 5.0\}$.

**Hyperparameter search for iSSD.** For iSSD, the edge budget $\eta$ in $\mathcal{L}_e$ is searched over: $\{0.5, 0.75, 0.85\}$; $K$ for the $K\%$ edges with lowest probability score for diversification is searched over:$\{50, 70, 90\}$; $\lambda_1$, $\lambda_2$ for balancing $\mathcal{L}_e$ and $\mathcal{L}_{div}$ are searched over: $\{10, 40\}$ and $\{1e-1, 1e-2, 1e-3\}$ respectively. The learnable data transformation function $t(\cdot)$ is searched over $\{GIN, GCN\}$, with the number of layers: $\{2, 3, 4\}$.

### H.3 MORE EXPERIMENTAL RESULTS

We provide more experiment details regarding: (1) Experiment results when there are multiple invariant substructures in a graph. (2) Effectiveness of iSSD in handling concept shift. (3) Experiment results for more application domains. (4) Ablation study on ERM pretraining. (5) The capability of iSSD of identifying spurious edges. (6) Hyperparameter sensitivity analysis on GOODHIV scaffold, OGBG-Molbbbp, and EC50 assay datasets, in Figure 6. (7) More visualization results on GOOD-Motif base and GOOD-Motif size in Figure 7 8.

**Model performance for graphs with multiple invariant subgraphs.** While Assumption 1 assumes the existence of a single invariant substructure causally related to each target label, many real-world graph applications (Hu et al., 2020; Gui et al., 2022) may contain multiple such invariant subgraphs. However, Assumption 1 can be reformulated to accommodate multiple $G_c$ without compromising the validity of our assumptions and theoretical results. Specifically, suppose there are $K$ invariant subgraphs, denoted as $G_{c,i}$ for $i \in [K]$. For any specific $G_{c,i}$, the spurious subgraph $G'_s$ can be redefined as $G'_s = G_s \cup \{G_{c,j} \mid j \neq i\}$. Given this redefinition, and under the presence of $G_s$, our assumption $I(G_{c,i}; Y) > I(G'_s; Y)$ holds for any $i \in [K]$. Consequently, the assumptions and theoretical results presented in this work remain valid, even when multiple $G_c$ exist within the datasets. To further support our claim, we curated a dataset based on SPMotif (Wu et al., 2022c), where in the train/valid/test datasets, two invariant substructures are attached to the spurious subgraph. Our method performs effectively under this scenario, as shown in Table 4.

**Effectiveness of iSSD in handling concept shift.** Intuitively, diversifying spurious subgraphs helps weaken the spurious correlations in the training data, thereby facilitating OOD generalization. To evaluate whether iSSD can indeed facilitate OOD generalization for datasets with concept shift, we

perform experiments on SPMotif datasets and GOOD-HIV dataset with size shift. The results are illustrated in Table 5.

Table 5: Model performance on datasets with concept shift.

| Method | SPMotif | | GOODHIV-Size |
|--------|---------|---------|--------------|
| | b=0.40 | b=0.60 | concept |
| ERM | $59.42_{\pm2.63}$ | $60.45_{\pm5.21}$ | 63.26 |
| IRM | $59.89_{\pm4.87}$ | $58.10_{\pm4.86}$ | 59.90 |
| Vrex | $61.16_{\pm3.06}$ | $56.88_{\pm1.19}$ | 60.23 |
| DisC | $57.03_{\pm10.42}$ | $51.28_{\pm9.46}$ | $72.69_{\pm1.64}$ |
| GSAT | $64.49_{\pm1.60}$ | $61.27_{\pm1.42}$ | $56.76_{\pm7.16}$ |
| GREA | $62.08_{\pm4.63}$ | $59.07_{\pm5.94}$ | $60.07_{\pm5.40}$ |
| CIGA | $65.23_{\pm3.58}$ | $62.17_{\pm2.28}$ | $73.62_{\pm0.86}$ |
| AIA | $65.11_{\pm2.47}$ | $59.46_{\pm6.23}$ | $74.21_{\pm1.81}$ |
| iSSD | $\mathbf{67.78}_{\pm\mathbf{3.98}}$ | $\mathbf{65.50}_{\pm\mathbf{3.53}}$ | $\mathbf{79.50}_{\pm\mathbf{1.57}}$ |

The results of ERM, IRM and VRex for GOODHIV-size are obtained from Gui et al. (2022). As shown in the table, our method achieves the best test performance, indicating that iSSD effectively handles concept shift through spurious subgraph diversification with graph size constraints.

**Experiment results on more application domains.** To further evaluate the effectiveness of iSSD across different application domains, we conduct experiments on GOOD-CMNIST (Gui et al., 2022) and Graph-Twitter (Socher et al., 2013; Yuan et al., 2022) datasets.

Table 6: Test performance on GOOD-CMNIST and Graph-Twitter datasets.

| Method | CMNIST | Graph-Twitter |
|--------|--------|---------------|
| ERM | $28.60_{\pm1.87}$ | $60.47_{\pm2.24}$ |
| IRM | $27.83_{\pm2.13}$ | $56.93_{\pm0.99}$ |
| Vrex | $28.48_{\pm2.87}$ | $57.54_{\pm0.93}$ |
| DisC | $24.99_{\pm1.78}$ | $48.61_{\pm8.86}$ |
| GSAT | $28.17_{\pm1.26}$ | $60.96_{\pm1.18}$ |
| GREA | $29.02_{\pm3.26}$ | $59.47_{\pm2.09}$ |
| CIGA | $32.22_{\pm2.67}$ | $62.31_{\pm1.63}$ |
| AIA | $\mathbf{36.37}_{\pm\mathbf{4.44}}$ | $61.10_{\pm0.47}$ |
| iSSD | $\underline{33.89}_{\pm1.65}$ | $\mathbf{63.37}_{\pm\mathbf{0.76}}$ |

As demonstrated in Table 6, iSSD also achieves superior performance in application domains beyond molecular applications, indicating its superior OOD performance and broad applicability.

**Ablation study on ERM pretraining.** We conduct ablation study across 5 datasets without using ERM pretraining. The results are presented in Table 7. As illustrated, incorporating ERM pretraining improves OOD performance in most cases, as the GNN encoder is able to learn useful representations before incorporating $L_e$ and $L_{div}$ to train $t(\cdot)$. Intuitively, this facilitates the optimization of $t(\cdot)$, therefore improving the test performance.

Table 7: Ablation study on test datasets.

| | Motif-basis | Motif-size | EC50-Assay | EC50-Sca | HIV-size |
|---|-------------|------------|------------|----------|----------|
| w/ pretrain | $91.48_{\pm0.40}$ | $66.53_{\pm8.55}$ | $78.01_{\pm0.42}$ | $67.56_{\pm1.63}$ | $64.99_{\pm1.63}$ |
| wo/ pretrain | $91.04_{\pm0.76}$ | $61.48_{\pm8.29}$ | $76.58_{\pm2.14}$ | $66.19_{\pm1.56}$ | $65.46_{\pm1.85}$ |

**The capability of iSSD of identifying spurious edges.** To verify the ability of iSSD to identify spurious edges while preserving critical edges in $G_c$, we conduct experiments and provide empirical

results on *Recall@K* and *Precision@K* on GOODMotif datasets. As illustrated in Table 8, iSSD is able to identify a subset of spurious edges with precision higher than $90\%$ across all datasets, even with $K = 50$, indicating that iSSD can preserve $G_c$ in the augmented graph samples, therefore improve the quality of generated graph samples.

Table 8: Recall@K and Precision@K for Motif-base and Motif-size datasets.

| K% | Motif-base | | Motif-size | |
|---|---|---|---|---|
| | Recall | Precision | Recall | Precision |
| 10% | 0.1467 | 1.0000 | 0.0963 | 0.9199 |
| 20% | 0.3076 | 0.9831 | 0.2023 | 0.9602 |
| 30% | 0.4556 | 0.9465 | 0.3093 | 0.9735 |
| 40% | 0.6056 | 0.9374 | 0.4153 | 0.9801 |
| 50% | 0.7356 | 0.9017 | 0.5243 | 0.9841 |

## H.4 SOFTWARE AND HARDWARE

We run all the experiments using PyTorch (Paszke et al., 2019) (version: 2.1.2) and PyTorch Geometric (Fey & Lenssen, 2019) (version: 2.4.0) on Linux servers with RTX 4090 and CUDA 11.8.

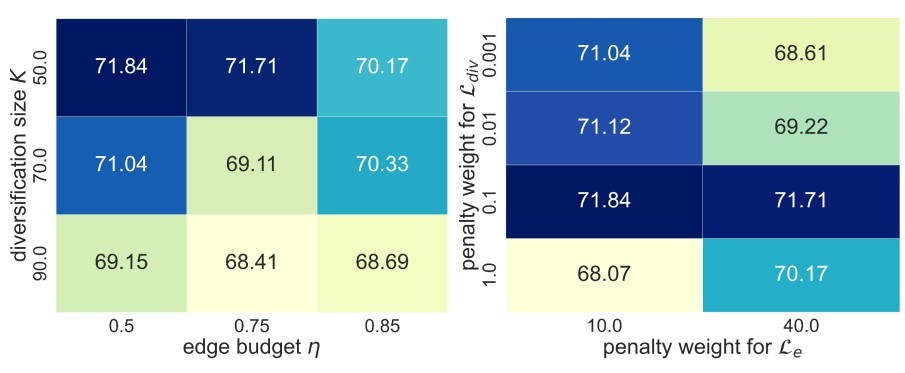

(a). Hyperparameter sensitivity on GOODHIV scaffold.

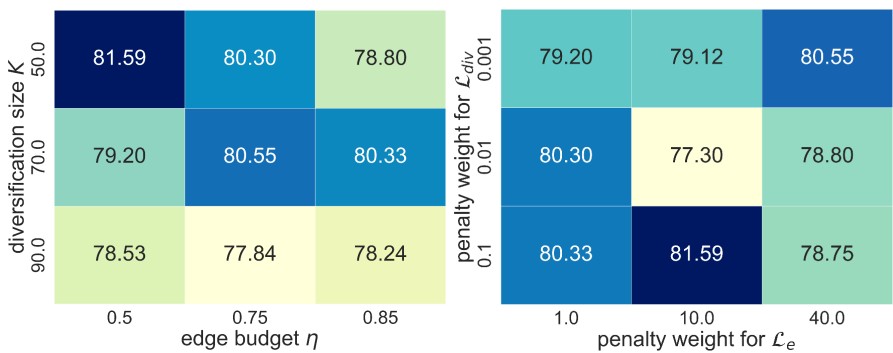

(b). Hyperparameter sensitivity on OGBG-Molbbbp size.

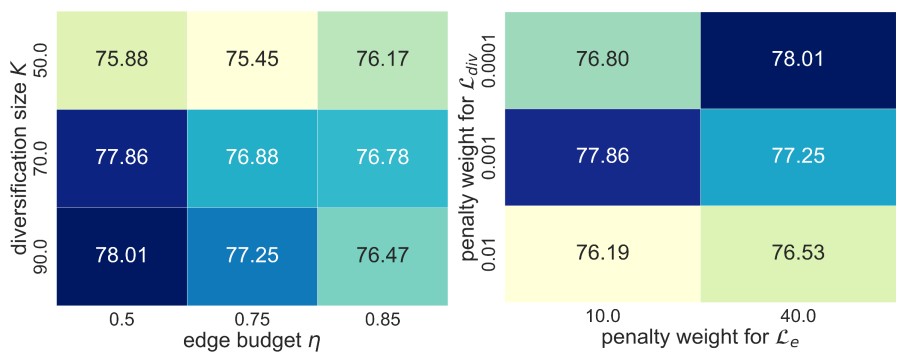

(c). Hyperparameter sensitivity on EC50 assay.

Figure 6: Hyperparameter sensitivity analysis across different datasets.

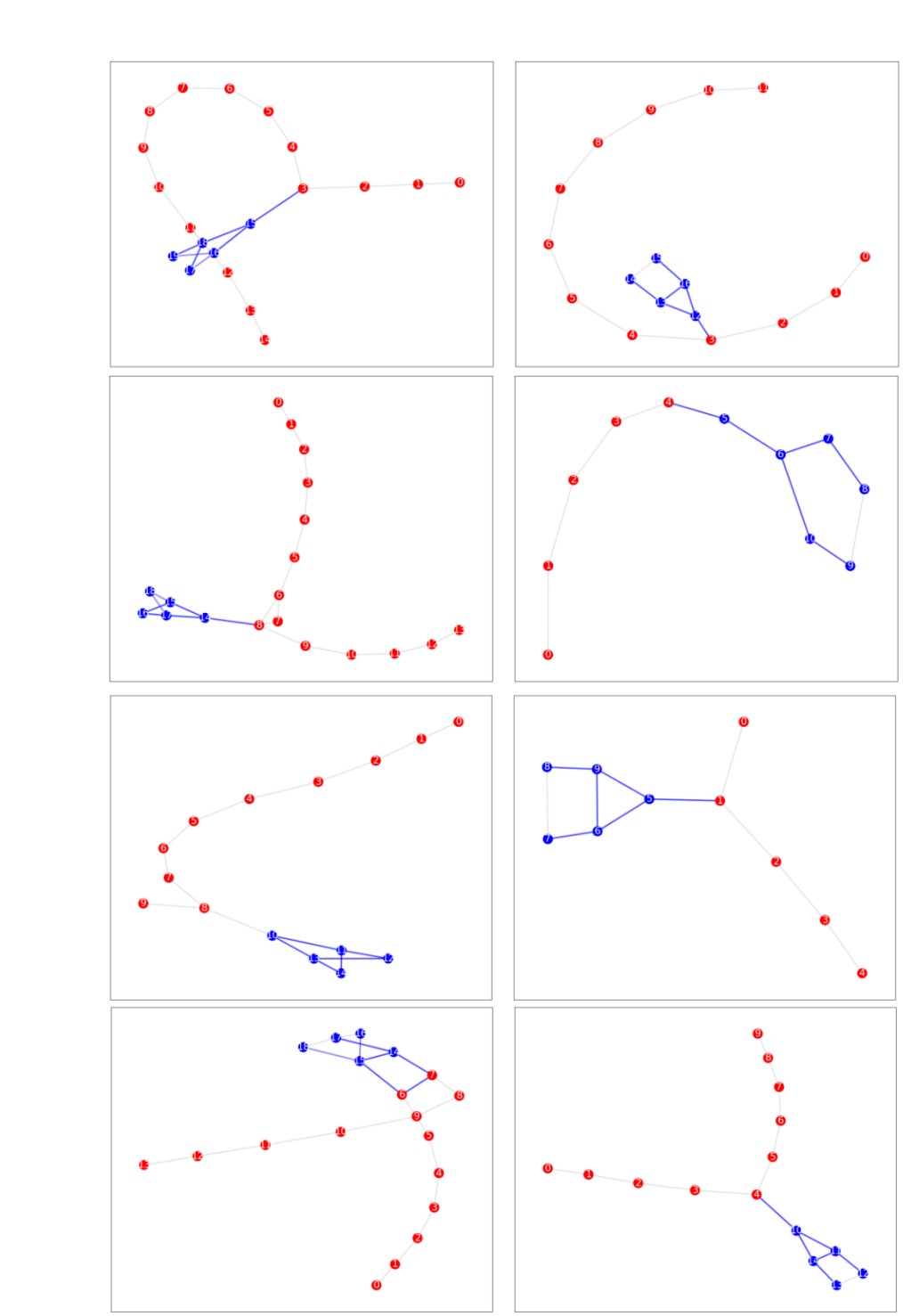

Figure 7: More visualization results on Motif-base dataset. The blue nodes are ground-truth nodes in $G_c$, and red nodes are ground-truth nodes in $G_s$. The highlighted blue edges are top-K edges predicted by $t^*(\cdot)$, where $K$ is the number of ground-truth causal edges in a graph.

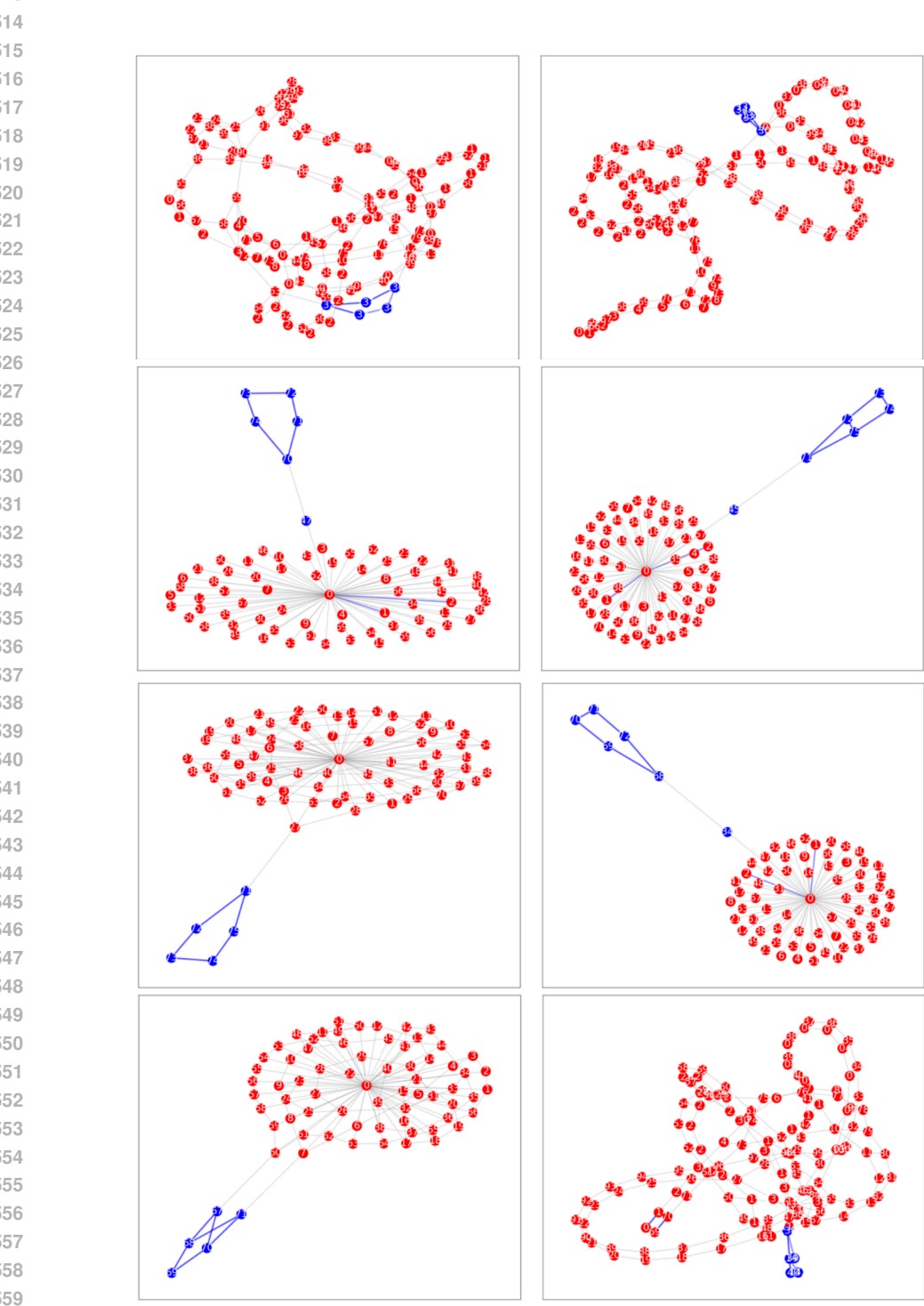

Figure 8: More visualization results on Motif-size dataset. The blue nodes are ground-truth nodes in $G_c$, and red nodes are ground-truth nodes in $G_s$. The highlighted blue edges are top-K edges predicted by $t^*(\cdot)$, where $K$ is the number of ground-truth causal edges in a graph.

