# OpenReview forum: "Diversifying Spurious Subgraphs for Graph Out-of-Distribution Generalization"
_ICLR.cc/2025/Conference — Submitted to ICLR 2025_

### Official Review · Reviewer_Jzm8 · 2024-10-24

**Soundness:** 2
**Presentation:** 3
**Contribution:** 2
**Rating:** 5
**Confidence:** 2

**Summary:**

Environment augmentation methods have shown promise in addressing out-of-distribution (OOD) generalization challenges in Graph Neural Networks (GNNs), but they face a trade-off between generating diverse graphs and preserving invariant substructures. Current methods are limited by deterministic graph composition strategies and the risk of excluding important subgraphs due to spurious correlations. To overcome these challenges, the authors propose a method called spurious subgraph diversification, which randomizes spurious subgraph generation while maintaining key invariant structures, achieving up to 24.19% better performance than 17 baseline methods on both synthetic and real-world datasets.

**Strengths:**

- The problem of graph OOD generalization is important.
- The framework provides detailed theoretical analysis, including bounds on generalization performance.
- The empirical results, involving comparisons with 17 baselines across multiple datasets, demonstrate the effectiveness of the method.
- The paper includes detailed ablation studies and sensitivity analyses to highlight the importance of each component of the iSSD framework.

**Weaknesses:**

- A major concern is about the soundness of the method "to idenitfy edges from spurious subgraphs accurately, we utilize the bottom K% of edges with the lowest predicted probabilities as estimated spurious edges". First, this identification of spurious subgraphs may not hold in many scenarios, where some spurious edges have strong correlations with the labels for the model to exploit, and thus for these edges model has high predicted probabilities. Second, in this way, there no supervision signals for capturing spurious edges, how do the authors guarantee that the captured edges are indeed spurious? Third, a surge of works in graph ood generalization propose different methods to capture spurious/variant subgraphs, how do this method surpass these methods?

- Another major concern is about the novelty of the method. Diversifying the spurious subgraphs, or environments, to improve graph ood generalization performance has been explored in many related works, e.g., EERM in ICLR22, which is not compared as baseline.  Also, the learnable data transformation module is similar to DIR in ICLR22.

- (minor) typo. In line 72,  "idenitfy".

**Questions:**

see weaknesses.

---

> ### Author Response · Authors · 2024-11-20
> **Reply to Reviewer Jzm8 (Part 1/2)**
>
> *We sincerely thank your valuable feedback and constructive suggestions! We are encouraged that you acknowledged the problem importance, detailed theoretical analysis and comprehensive experiments. Please see below for our responses to your comments and concerns.*
>
> ---
>
> >**W1: Soundness of the method "to idenitfy edges from spurious subgraphs accurately, we utilize the bottom K% of edges with the lowest predicted probabilities as estimated spurious edges".**
>
>
> **Responses:** Thank you for your insightful comments!
>
> Some spurious edges have strong correlations with the labels for the model to exploit, and thus for these edges model has high predicted probabilities.
>
> - We agree with the reviewer that such cases exist in some scenarios, and some spurious edges may indeed exhibit high predicted probabilities. However, we respectively claim that in such cases our method can still accurately identify (a subset of) the spurious edges, due to the following main reasons.
>   - First, our method is ranking-based. Therefore, even if a few spurious edges have stronger correlation with the labels and exhibit high probabilities, utilizing the bottom K% edges with the lowest prediction probability can still identify a subset of spurious edges. Moreover, as edges in $G_c$ tend to also exhibit high probabilities (Prop.2), the bottom K% edges won't include edges in $G_c$ with high probability given a suitable $K$.
>   - Second, the theoretical guarantee in Proposition 2 only requires the mutual information $I(G_C; Y)$ is greater than $I(G_S; Y)$, in which both measures are defined on the population-level, not edges-level. That is, Proposition 2 holds as long as most of (rather than all) the invariant edges have stronger correlations with the labels compared to the spurious edges.
>
> - To verify the effectiveness of our method for capturing spurious edges while preversing edges in $G_c$ when some spurious edges have strong correlations, we have included more experiments on the GOODMotif datasets, with the following discussions.
>   - For the Motif-base dataset, where no strong correlation exists in $G_s$, iSSD achieves superior performance, comparable to in-distribution ERM performance.
>   - However, for the Motif-size dataset, the increased topological complexity in $G_s$ results in some spurious substructures having strong correlations with the targets. Consequently, as the reviewer pointed out, our method cannot capture all spurious edges. Nevertheless, we can still identify a subset of these edges, which explains why our method can still achieve ~66% accuracy in this more challenging scenario, outperforming all other baselines.
>   - To verify the capability of capturing spurious edges while preserving stable edges, we added  more experimental results on _Recall_ and _Precision_ for the GOODMotif datasets on $K\\%$ lowest probability edges predicted by  $t(\cdot)$.
>
> __Table: _Recall_ and _Precision_ for the Motif-base dataset__
> | **K\%**  | **Recall** | **Precision** |
> |--------|---------------------|-----------------------|
> | 10\%     | 0.1467             | 1.0000               |
> | 20\%     | 0.3076             | 0.9831               |
> | 30\%     | 0.4556             | 0.9465               |
> | 40\%     | 0.6056             | 0.9374               |
> | 50\%     | 0.7356             | 0.9017               |
>
>
> __Table: _Recall_ and _Precision_ for the Motif-size dataset__
>
> | **K\%**  | **Recall** | **Precision** |
> |--------|---------------------|-----------------------|
> | 10\%     | 0.0963             | 0.9199               |
> | 20\%     | 0.2023             | 0.9602               |
> | 30\%     | 0.3093             | 0.9735               |
> | 40\%     | 0.4153             | 0.9801               |
> | 50\%     | 0.5243             | 0.9841               |
>
> As illustrated, on the more challenging Motif-size dataset, our method can stably identify a subset of spurious edges with precision higher than $90\\%$ even with bottom $50\\%$ edges. Moreover, our method outperforms other baseline methods by a large margin even when only a subset of spurious edges is identified.

---

> ### Author Response · Authors · 2024-11-20
> **Reply to Reviewer Jzm8 (Part 2/2)**
>
> #### **How does iSSD surpasses existing method in terms of capturing spurious/variant subgraphs?**
>
> - To clarify, we provide a novel perspective and approach of removing spurious edges while preserving $G_c$. Specifically, previous methods attempt to directly identify $G_c$ to enhance OOD generalization ability, but this approach can be error-prone, especially when some spurious edges are highly correlated with the targets and exhibit high probability. In contrast, our ranking-based approach prunes a subset of spurious edges, therefore ensure $G_c$ can be correctly preserved, which is at the cost of including a portion of high-probability spurious edges that are challenging to be get rid of.
> - Previous methods often rely on deterministic graph composition strategies to combine estimated invariant and spurious subgraphs, which can limit the diversity of the generated graphs. To address this, we diversify the  spurious edges for  the generation of spurious subgraphs and producing diverse spurious patterns. Our method promotes diversity while ensuring the preservation of $G_c$, thus facilitates the model’s ability to extrapolate to unseen environments effectively.
>
> > **W2: Novelty of the method.**
>
> **Responses:** Thank you for your insightful comments, we would like to clarify the main differences between our method and the mentioned previous methods as follows.
>
> #### **Comparison to EERM**
>
> - EERM indeed employs multiple auxiliary context generators to mimic diverse training environments from the training data. However, these generators may still suffer from strong spurious correlations and generate inaccurate environments due to their objective of maximizing the variance loss with model-feedback, which is somewhat similar to AIA.
> - In contrast, our approach directly randomizes the lowest $K\\%$ edges for environment extrapolation. By relying on a ranking-based approach, the issues caused by spurious correlations can be mitigated through an appropriate choice of $K$.
> - Additionally, since EERM primarily focuses on node-level OOD generalization, and given the unique challenges and differing task characteristics, we primarily compare our method with graph-level invariant learning and data augmentation methods, such as AIA, which also aims to learn diverse environments using an adversarial augmentation generator.
>
>
> #### **Comparison to DIR**
>
> In Section 5, we provide a detailed discussion on how our approach differs from prior studies, despite the use of a learnable data transformation module.
>
> - In brief, although the concept of a learnable data transformation is not novel and is not our contribution, our work identifies potential limitations in terms of the quality of augmented graph in previous studies which adopt the graph composition strategies with restrictive form (see Appendix G for details).
> - Motivated by this observation, we introduce a new perspective on how to mitigate these limitations for more effective environment extrapolation.
> - Concretely, our method discards any graph composition strategy with restrictive form, and introduce spurious subgraph diversification with a graph size constraint.
> - Our method significantly simplifies the process compared to previous approaches while maximally extrapolate to novel environments, therefore enhancing the OOD generalization ability.
>
> > **W3: Minor typos**
>
> **Responses:** Thank you for your careful review. We have thoroughly proofread the draft and corrected the identified typos, with changes highlighted in blue.
>
> ---
>
> We sincerely thank you for your careful review and approval of the effectiveness and theoretical analysis. We hope we have addressed your concerns on the novelty and soundness of our approach.
>
> ---
>
> __References__
>
> [1] Wu, et al., Discovering invariant rationales for graph neural networks, ICLR 2022.
>
> [2] Chen, et al., Learning causally invariant representations for out-of-distribution generalization on graphs, NeurIPS 2022.
>
> [3] Li, et al., Learning invariant graph representations for outof-distribution generalization, NeurIPS 2022.
>
> [4] Liu, et al., Graph rationalization with environment-based augmentations. KDD2022.
>
> [5] Sui, et al., Unleashing the Power of Graph Data Augmentation on Covariate Distribution Shift. NeurIPS 2023.
>
> [6] Gui, et al., Joint Learning of Label and Environment Causal Independence for Graph Out-of-Distribution Generalization. NeurIPS 2023.

---

> ### Author Response · Authors · 2024-11-25
>
> Dear Reviewer Jzm8,
>
> As the discussion deadline approaches, we are wondering whether our responses have properly addressed your concerns regarding the soundness and novelty of our work? Your feedback would be extremely helpful to us. If you have further comments or questions, we hope for the opportunity to respond to them.
>
> Many thanks,
>
> Authors

---

> ### Author Response · Authors · 2024-12-02
> **Gentle reminder**
>
> Once again, we are grateful for your time and effort for reviewing our paper. Since the discussion period will end in a few days, we are very eager to get your feedback on our response. We understand that you are very busy, but we would highly appreciate it if you could take into account our response and reconsider your evaluation if some concerns are addressed. Thank you!
>
>
> Authors

---

### Official Review · Reviewer_CgQ7 · 2024-10-28

**Soundness:** 3
**Presentation:** 3
**Contribution:** 3
**Rating:** 6
**Confidence:** 3

**Summary:**

The paper proposes a novel environment augmentation method for graph out-of-distribution (OOD) generalization. Input graphs are assumed to be composed of an invariant/causal subgraph $G_c$ that predicts the target and a spurious subgraph $G_s$ that encodes environment variability. Their approach involves learning a data transformation with high sampling probability for edges in $G_c$ and a low sampling probability for edges in $G_s$. This is done via a combination of cross-entropy loss and two new regularization terms: (1) Graph size constraint $L_e$ to control the number of pruned edges during transformation; (2) Spurious subgraph diversification $L_{div}$ to maximize randomness in the spurious subgraphs retained after transformation. This is supplemented by theoretical proofs showing that $L_e$ tightens OOD generalization bound while $L_{div}$ improves OOD generalization. To empirically verify these claims, they benchmark performance against 17 baseline methods across 4 dataset classes, perform ablation studies on the regularization terms, evaluate sensitivity to hyper-parameters, and other in-depth analysis.

**Strengths:**

1. The paper is well-written and all claims and design choices are well-motivated with appropriate references to literature.
2. It presents a novel idea grounded in theory to tackle OOD graph generalization, a problem of interest for broader graph community
3. Along with performance comparison against baseline methods, they include sufficient empirical analysis to justify their design choices and insights to support theoretical claims.

**Weaknesses:**

1. Limited data diversity: Apart from the synthetic GOODMotif dataset, all others pertain to molecules. In these datasets, it may be possible to ascribe target graph label to specific functional groups and thus, ascertain the invariant subgraphs. While it is alright to limit the scope of experiments, I wonder how the proposed method performs against datasets from other application domains (CMNIST, GOOD-Arxiv). It may be useful to provide more insight or comment on the broader applicability of proposed approach.

**Questions:**

1. Do we have an intuition on why the performance drops for Motif-Base dataset on going from K=90 to 70 and rises again for K=50 for all $\eta$ in Figure 2?
2.  Do we have an insight into the cause of higher variance on removing either $L_e$ or $L_{div}$?
3. Why was ERM pre-training used? Was it used with other baseline methods as well? If not, why? Could we include performance w/o ERM training to isolate and analyze performance gains from proposed techniques?
4. Both the GOOD datasets use covariate shift split. Is the method applicable to concept shift?
5. How restrictive is the regularization from $L_e$? Do we know how the number of edges are distributed in graphs sampled from $t(G)$?

---

> ### Author Response · Authors · 2024-11-20
> **Reply to Reviewer CgQ7 (Part 1/4)**
>
> *We sincerely thank you for your valuable feedback and constructive suggestions. We are encouraged that you acknowledged we propose a novel idea to tackle OOD graph generalization, with sufficient empirical analysis to justify their design choices and insights to support theoretical claims. Please see below for our responses to your comments and concerns.*
>
> >**W1: Limited data diversity**
>
> **Responses:** Thank you for the great suggestion! We have added the GOOD-CMNIST and Graph-Twitter datasets, which come from different application domains to enhance data diversity.
>
> - **GOOD-CMNIST:** This dataset is based on a modified version of the classic MNIST dataset, where digits are represented as graphs with nodes and edges. We adopt the covariate shift setting for this dataset.
>
> - **Graph-Twitter:** This dataset converts sentiment sentence classification datasets SST-Twitter into graphs, the split is based on degree shift.
>
> The results are presented below:
>
> **Table: Experimental results on the GOOD-CMNIST and Graph-Twitter datasets**
> | Method | GOOD-CMNIST | Graph-Twitter |
> |:---:|:---:|:---:|
> | ERM | 28.60±1.87 | 60.47±2.24 |
> | IRM | 27.83±2.13 | 56.93±0.99 |
> | Vrex | 28.48±2.87 | 57.54±0.93 |
> | DisC | 24.99±1.78 | 48.61±8.86 |
> | GSAT | 28.17±1.26 | 60.96±1.18 |
> | GREA | 29.02±3.26 | 59.47±2.09 |
> | CIGA | 32.22±2.67 | 62.31±1.63 |
> | AIA | **36.37±4.44** | 61.10±0.47 |
> | iSSD (Ours) | 33.89±1.65 | **63.37±0.76** |
>
> As demonstrated, iSSD also achieves superior performance in application domains beyond molecular applications, this hightlight the broad applicability of our proposed method.
>
> We have added these additional results in Appendix H.3 in our revised draft to improve our paper quality.
>
> > **Q1: performance drop and rise with different $\eta$ and $K$**
>
> **Responses:** This is a great observation! We reran the experiments on the Motif-Base dataset with different random seeds. For $K=70$ and $\eta=0.85$, the accuracy reached an average of around 86% over 3 runs and around $72\%$ for $\eta=0.75$. So it may be caused by the random initialization. Since the model need to sort based on the lowest probability edges and then diversify on these edges, the randomness can introduce some uncertainty, thereby affecting the final results. However, as a general guideline, selecting an $\eta$ that is not too small, such as 0.75 or 0.85, and choosing penalty weights for $\mathcal{L}_{div}$ that are not too large, such as 1e-2 or 1e-3 is usually good for various datasets.
>
> > **Q2: Reason that $\mathcal{L}\_{div}$ and $\mathcal{L}\_{e}$ cause high variance**
>
> **Responses:** Please see our response below:
>
> - As $\mathcal{L}\_{div}$ maximally randomizes the spurious subgraphs by setting the edge probability to $0.5$, it generates a variety of novel spurious patterns for environment extrapolation. However, this approach also introduces substantial randomness by diversifying the spurious subgraphs, which can lead to higher variance.
> - Regarding $\mathcal{L}\_{e}$, without the diversification, the model struggles to extrapolate effectively to unseen environments. Consequently, in test datasets with novel environments, the model may inadvertently prune edges in $G_c$, leading to performance instability.
>
> When iSSD combines $\mathcal{L}\_{div}$ and $\mathcal{L}\_{e}$, the model can effectively prune spurious edges, and extrapolate to unseen environments, therefore reducing variance on test dataset. One such example is Motif-base dataset, where the spurious edges are removed, while causal edges are preversed, therefore the variance is very small. In contrast, for Motif-size dataset, as there exists some edges in $G_s$ cannot be pruned due to the large size and topological complexity, which leads to high variance in this dataset.
>
> > **Q3: Rationale for using ERM pretraining and associated ablation study.**
>
> **Responses:** Thank you for this insightful question. The reason for using ERM pretraining to train the GNN encoder is to allow the encoder to first learn useful representations before incorporating $L_e$ and $L_{div}$ to train $t(\cdot)$. Intuitively, this facilitates the optimization of $t(\cdot)$. Similarly, CIGA also adopts ERM pretraining, and GSAT gradually decays $r$ (the prior for the edge distribution) from 0.9 to a fixed value, which is conceptually similar to ERM pretraining.
>
> While other baseline methods do not explicitly use ERM pretraining, they incorporate techniques unique to their methods to achieve better performance. For example, AIA employs _CosineAnnealingLR_ as the learning rate scheduler and only evaluates on the test set after certain epochs (e.g., after 70 epochs). Similarly, GREA also adopts _CosineAnnealingLR_ and trains for up to 400 epochs, as reported in their papers.
>
> To further analyze the impact of ERM pretraining, we added an ablation study across 5 datasets without using ERM pretraining. The results are presented in the following table:

---

> ### Author Response · Authors · 2024-11-20
> **Reply to Reviewer CgQ7 (Part 2/4)**
>
> **Table: Ablation study on ERM pretraining**
> |  | Motif-basis | Motif-size | EC50-Assay | EC50-Sca | HIV-size |
> |---|---|---|---|---|---|
> | w/ pretrain | 91.48±0.40 | 66.53±8.55 | 78.01±0.42 | 67.56±1.63 | 64.99±1.63 |
> | w/o pretrain | 91.04±0.76 | 61.48±8.29 | 76.58±2.14 | 66.19±1.56 | 65.46±1.85 |
>
> As we can see from the results, incorporating ERM pretraining improves OOD performance in most cases. However, for larger datasets such as GOODHIV-size, the improvement is less significant and may not be as helpful.
>
> We have added these additional results in Appendix H.3 in our revised draft and highlight it in blue.
>
> > **Q4: Performance on datasets with concept shift.**
>
> **Responses:** Intuitively, diversifying spurious subgraphs helps weaken the spurious correlations in the training data, thereby facilitating OOD generalization. To address the reviewer's concern, we have added experiments on  3 datasets with _concept shift_, as demonstrated in the following table:
>
> **Table: Experimental results on addressing concept shift**
> | Method | SPMotif (b=0.40) | SPMotif (b=0.70) | HIV-size (concept) |
> |:---:|:---:|:---:|:---:|
> | ERM | 59.42±2.63 | 60.45±5.21 | 63.26 |
> | IRM | 59.89±4.87 | 58.10±4.86 | 59.90 |
> | Vrex | 61.16±3.06 | 56.88±1.19 | 60.23 |
> | DisC | 57.03±10.42 | 51.28±9.46 | 72.69±1.64 |
> | GSAT | 64.49±1.60 | 61.27±1.42 | 56.76±7.16 |
> | GREA | 62.08±4.63 | 59.07±5.94 | 60.07±5.40 |
> | CIGA | 65.23±3.58 | 62.17±2.28 | 73.62±0.86 |
> | AIA | 65.11±2.47 | 59.46±6.23 | 74.21±1.81 |
> | iSSD (Ours) | **67.78±3.98** | **65.50±3.53** | **79.50±1.57** |
>
> The results of ERM,IRM and VRex for GOODHIV-size are obtained from [1]. As shown in the table, our method achieves the best test performance, indicating that iSSD effectively handles _concept shift_ through spurious subgraph diversification combined with graph size constraints.
>
> We have added these additional results in Appendix H.3 in our revised draft and highlighted it in blue.
>
> > **Q5: How restrictive is $\mathcal{L}_e$, and the edge distribution from $t(G)$.**
>
> **Responses:** $\mathcal{L}_e$ is primarily used to encourage $t(\cdot)$ to induce sparsity in terms of the total number of edges. In practice, our hyperparameter analysis on the Motif-base dataset (Figure 2) and OGBG-Molbbbp dataset (Figure 6) indicates that the penalty for $L_e$ should be relatively large (e.g., 10 or 40) to achieve optimal performance.
>
> To address the reviewer's question regarding the edge distribution sampled from $t(G)$, we further analyzed the distribution of edges on the GOOD-Motif dataset. We have the following observations:
>
>   - In the Motif-base dataset, most spurious edges are driven down to near zero probability, while all causal edges in $G_c$ exhibit high probabilities as predicted by $t(\cdot)$. This explains why iSSD performs almost identically to in-distribution ERM on the Motif-base dataset.
>   - In the Motif-size dataset, the problem is more challenging because $G_s$ is significantly larger than $G_c$, and the topology of $G_s$ is more complex compared to Motif-base. As a result, many substructures within $G_s$ may resemble those in $G_c$. While $t(G)$ predicts high probabilities for most causal edges in Motif-size dataset, some spurious edges also have high probabilities of being sampled by $t(G)$, which negatively impacts OOD performance. Nevertheless, while most OOD methods perform similarly to ERM, iSSD significantly outperforms ERM and state-of-the-art OOD methods by a large margin in Motif-size dataset.
>
> We also provide 2 examples from each dataset to offer a more intuitive understanding of these observations as follows:
>
> ```
> The first tensor indicates the edge probability predicted by t(G), the second tensor is the ground-truth for the edges.
>
> For Motif-base dataset,
>
> # graph-1
>
> tensor([2.27e-18, 3.79e-03, 2.27e-18, 1.49e-23, 1.49e-23, 1.49e-23,
>  1.49e-23, 1.49e-23, 1.49e-23, 7.72e-27, 7.72e-27, 1.73e-36,
>  1.73e-36, 3.79e-03, 9.99e-01, 9.99e-01, 9.99e-01, 4.80e-01,
>  4.80e-01, 4.80e-01, 4.80e-01, 9.99e-01, 4.80e-01, 4.80e-01,
>  4.80e-01, 4.80e-01] )
> tensor([False,  True, False, False, False, False, False, False, False, False,
>         False, False, False,  True,  True,  True,  True,  True,  True,  True,
>          True,  True,  True,  True,  True,  True])
>
> # graph-2
>
> tensor([1.73e-36, 1.73e-36, 1.17e-21, 1.17e-21, 9.38e-05, 9.38e-05,
>     9.38e-05, 1.00e+00, 9.38e-05, 2.27e-18, 2.27e-18, 1.49e-23,
>     1.49e-23, 1.49e-23, 1.49e-23, 1.49e-23, 1.49e-23, 1.49e-23,
>     1.49e-23, 7.72e-27, 7.72e-27, 1.73e-36, 1.73e-36, 1.00e+00,
>     1.00e+00, 1.00e+00, 1.00e+00, 3.79e-03, 1.00e+00, 3.79e-03,
>     3.32e-13, 3.32e-13, 3.79e-03, 1.00e+00, 1.00e+00, 3.79e-03])
> tensor([False, False, False, False, False, False, False, False, False, False,
>         False, False, False, False, False, False, False, False, False, False,
>         False, False, False, False,  True,  True,  True,  True,  True,  True,
>          True,  True,  True,  True,  True,  True])
> ```

---

> ### Author Response · Authors · 2024-11-20
> **Reply to Reviewer CgQ7 (Part 3/4)**
>
> ```
> For Motif-size dataset,
>
> # graph-1
>
> tensor([9.99e-01, 9.99e-01, 9.99e-01, 1.00e+00, 1.00e+00, 9.99e-01,
>         1.00e+00, 1.00e+00, 1.00e+00, 1.00e+00, 1.00e+00, 1.00e+00,
>         1.00e+00, 1.00e+00, 1.00e+00, 1.0000e+00, 1.0000e+00, 1.0000e+00,
>         1.0000e+00, 1.0000e+00, 1.0000e+00, 6.2184e-02, 6.2184e-02, 1.0000e+00,
>         1.1312e-02, 9.8619e-01, 1.0000e+00, 1.0000e+00, 6.2184e-02, 6.2184e-02,
>         1.0000e+00, 9.9632e-01, 6.2184e-02, 1.0000e+00, 6.2184e-02, 9.9632e-01,
>         1.0000e+00, 6.2184e-02, 6.2184e-02, 1.0000e+00, 9.9632e-01, 9.9632e-01,
>         1.0000e+00, 6.2184e-02, 6.2184e-02, 6.2184e-02, 4.3103e-06, 4.3103e-06,
>         6.2184e-02, 4.3103e-06, 4.3103e-06, 1.1312e-02, 2.8521e-07, 2.8521e-07,
>         8.2258e-02, 9.8619e-01, 5.2499e-01, 1.8020e-02, 6.2184e-02, 4.3103e-06,
>         4.3103e-06, 6.2184e-02, 4.3103e-06, 4.3103e-06, 9.9632e-01, 2.1233e-02,
>         7.8001e-01, 6.2184e-02, 4.3103e-06, 4.3103e-06, 6.2184e-02, 4.3103e-06,
>         4.3103e-06, 9.9632e-01, 2.1233e-02, 5.4943e-01, 6.2184e-02, 4.3103e-06,
>         4.3103e-06, 6.2184e-02, 4.3103e-06, 4.3103e-06, 9.9632e-01, 2.1233e-02,
>         7.8472e-01, 9.9632e-01, 7.8472e-01, 2.1233e-02, 6.2184e-02, 4.3103e-06,
>         4.3103e-06, 6.2184e-02, 4.3103e-06, 4.3103e-06, 4.3103e-06, 4.3103e-06,
>         4.3103e-06, 4.3103e-06, 2.8521e-07, 2.8521e-07, 5.2499e-01, 7.0776e-01,
>         1.8020e-02, 4.3103e-06, 4.3103e-06, 4.3103e-06, 4.3103e-06, 2.1233e-02,
>         7.8001e-01, 9.9999e-01, 4.3103e-06, 4.3103e-06, 4.3103e-06, 4.3103e-06,
>         2.1233e-02, 5.4943e-01, 7.0776e-01, 4.3103e-06, 4.3103e-06, 4.3103e-06,
>         4.3103e-06, 2.1233e-02, 1.0000e+00, 7.8472e-01, 8.2258e-02, 7.8472e-01,
>         2.1233e-02, 4.3103e-06, 4.3103e-06, 4.3103e-06, 4.3103e-06, 9.9999e-01,
>         1.0000e+00, 1.0000e+00, 1.0000e+00, 9.9961e-01, 9.9961e-01, 9.9961e-01,
>         9.9961e-01, 1.0000e+00, 9.9961e-01, 9.9961e-01, 9.9961e-01, 9.9961e-01])
>
> tensor([False, False, False, False, False, False, False, False, False, False,
>         False, False, False, False, False, False, False, False, False, False,
>         False, False, False, False, False, False, False, False, False, False,
>         False, False, False, False, False, False, False, False, False, False,
>         False, False, False, False, False, False, False, False, False, False,
>         False, False, False, False, False, False, False, False, False, False,
>         False, False, False, False, False, False, False, False, False, False,
>         False, False, False, False, False, False, False, False, False, False,
>         False, False, False, False, False, False, False, False, False, False,
>         False, False, False, False, False, False, False, False, False, False,
>         False, False, False, False, False, False, False, False, False, False,
>         False, False, False, False, False, False, False, False, False, False,
>         False, False, False, False, False, False, False, False, False, False,
>         False, False,  True,  True,  True,  True,  True,  True,  True,  True,
>          True,  True,  True,  True] )
>
> ```

---

> ### Author Response · Authors · 2024-11-20
> **Reply to Reviewer CgQ7 (Part 4/4)**
>
> ```
> # graph-2
>
> tensor([2.5662e-07, 2.5662e-07, 9.7500e-03, 9.7500e-03, 4.0217e-01, 4.0217e-01,
>         4.0217e-01, 4.0217e-01, 4.0217e-01, 4.0217e-01, 4.0217e-01, 4.0217e-01,
>         4.0217e-01, 4.0217e-01, 4.0217e-01, 4.0217e-01, 4.0217e-01, 4.0217e-01,
>         4.0217e-01, 4.0217e-01, 4.0217e-01, 4.0217e-01, 4.0217e-01, 4.0217e-01,
>         4.9852e-01, 4.9852e-01, 9.8725e-01, 9.8725e-01, 9.8725e-01, 1.0000e+00,
>         1.0000e+00, 9.8725e-01, 4.9852e-01, 4.9852e-01, 4.0217e-01, 4.0217e-01,
>         4.0217e-01, 4.0217e-01, 4.0217e-01, 4.0217e-01, 4.0217e-01, 4.0217e-01,
>         4.0217e-01, 4.0217e-01, 4.0217e-01, 4.0217e-01, 4.0217e-01, 4.0217e-01,
>         4.0217e-01, 4.0217e-01, 4.0217e-01, 4.0217e-01, 4.0217e-01, 4.0217e-01,
>         4.0217e-01, 4.0217e-01, 4.0217e-01, 4.0217e-01, 4.0217e-01, 4.0217e-01,
>         4.9852e-01, 4.9852e-01, 9.6026e-01, 9.6026e-01, 9.6026e-01, 1.0000e+00,
>         9.6026e-01, 4.9852e-01, 4.9852e-01, 4.0217e-01, 4.0217e-01, 4.0217e-01,
>         4.0217e-01, 4.0217e-01, 4.0217e-01, 4.0217e-01, 4.0217e-01, 4.0217e-01,
>         4.0217e-01, 4.0217e-01, 4.0217e-01, 4.0217e-01, 4.0217e-01, 4.0217e-01,
>         4.0217e-01, 4.0217e-01, 4.0217e-01, 4.9852e-01, 4.9852e-01, 9.6378e-01,
>         1.0000e+00, 9.6378e-01, 9.6378e-01, 9.6378e-01, 4.9852e-01, 4.9852e-01,
>         4.0217e-01, 4.0217e-01, 4.0217e-01, 4.0217e-01, 4.0217e-01, 4.0217e-01,
>         4.0217e-01, 4.0217e-01, 4.0217e-01, 4.0217e-01, 4.0217e-01, 4.0217e-01,
>         4.0217e-01, 4.0217e-01, 4.0217e-01, 4.0217e-01, 4.0217e-01, 4.0217e-01,
>         4.9852e-01, 4.9852e-01, 9.6026e-01, 9.6026e-01, 9.6026e-01, 1.0000e+00,
>         9.6026e-01, 4.9852e-01, 4.9852e-01, 4.0217e-01, 4.0217e-01, 4.0217e-01,
>         4.0217e-01, 4.0217e-01, 4.0217e-01, 4.0217e-01, 4.0217e-01, 4.0217e-01,
>         4.0217e-01, 4.0217e-01, 4.0217e-01, 4.0217e-01, 4.0217e-01, 4.0217e-01,
>         4.0217e-01, 4.0217e-01, 4.0217e-01, 4.0217e-01, 4.0217e-01, 4.0217e-01,
>         4.0217e-01, 9.7500e-03, 9.7500e-03, 2.5662e-07, 2.5662e-07, 1.0000e+00,
>         9.9999e-01, 1.0000e+00, 9.9999e-01, 9.9961e-01, 1.0000e+00, 9.9961e-01,
>         9.9961e-01, 9.9961e-01, 9.9999e-01, 1.0000e+00, 1.0000e+00, 9.9999e-01])
>
> tensor([False, False, False, False, False, False, False, False, False, False,
>         False, False, False, False, False, False, False, False, False, False,
>         False, False, False, False, False, False, False, False, False, False,
>         False, False, False, False, False, False, False, False, False, False,
>         False, False, False, False, False, False, False, False, False, False,
>         False, False, False, False, False, False, False, False, False, False,
>         False, False, False, False, False, False, False, False, False, False,
>         False, False, False, False, False, False, False, False, False, False,
>         False, False, False, False, False, False, False, False, False, False,
>         False, False, False, False, False, False, False, False, False, False,
>         False, False, False, False, False, False, False, False, False, False,
>         False, False, False, False, False, False, False, False, False, False,
>         False, False, False, False, False, False, False, False, False, False,
>         False, False, False, False, False, False, False, False, False, False,
>         False, False, False, False, False, False, False, False, False, False,
>          True,  True,  True,  True, False,  True,  True,  True,  True, False,
>          True,  True])
> ```
>
> ---
>
> We are grateful for your positive evaluation of our well-motivated design choice, presentation, novelty, and  effectiveness. We hope our responses address your concerns. We appreciate your support and we are happy to engage further if there are any further questions!
>
>
> ---
>
>
> __Reference__
>
>
> [1] Gui, et al., GOOD: A Graph Out-of-Distribution Benchmark, NeurIPS2022.

---

> ### Author Response · Authors · 2024-11-25
>
> Dear Reviewer CgQ7,
>
> As the discussion deadline approaches, we are wondering whether our responses have properly addressed your concerns and questions? Your feedback would be extremely helpful to us. If you have further comments or questions, we hope for the opportunity to respond to them.
>
> Many thanks,
>
> Authors

---

### Official Review · Reviewer_Ww8s · 2024-10-31

**Soundness:** 3
**Presentation:** 4
**Contribution:** 4
**Rating:** 6
**Confidence:** 4

**Summary:**

This paper proposes a novel augmentation strategy for out-of-distribution (OOD) generalization by diversifying spurious subgraphs. The method, termed spurious subgraph diversification, employs randomized spurious subgraph generation to maximize diversity while preserving invariant substructures. This approach achieves state-of-the-art results on both synthetic and real-world datasets.

**Strengths:**

1. The authors provide a unified theoretical framework for Learnable Data Transformation under graph OOD scenarios, demonstrating a solid theoretical foundation.
2. The proposed iSSD framework is easy but effective. Extensive experimental results demonstrate the superiority of proposed method.
3. The paper is well-written, and easy to follow. I really enjoy reading this paper.

**Weaknesses:**

1. The authors propose a graph size constraint loss (Eq. 7) to make sure the model will prune edges from the spurious subgraphs. However, I am puzzled as to how this is guaranteed? Because this loss can only ensure that the model will remove edges according to the budget, it is entirely possible to minimize the Eq. 7 loss by removing the edges included in the causal subgraph.
2. I believe that the joint selection of $K$ and $\eta$ is tricky in the motif dataset. Additionally, for real-world datasets or practical model deployments, how to choose $K$ and $\eta$ without prior knowledge about the scale of "spurious subgraph"?
3. As this study focuses on augmentation-based methods for graph OOD generalization, it would be better that the authors to include additional baseline or more discussion about augmentation-based graph OOD generalization methods, such as [1-2].

[1] Graph Out-of-Distribution Generalization with Controllable Data Augmentation. TKDE 2024.

[2] Graph Invariant Learning with Subgraph Co-mixup for Out-Of-Distribution Generalization. AAAI 2024.

**Questions:**

Please address my concern in weakness part.

---

> ### Comment · Reviewer_Ww8s · 2024-11-14
> **Some questions at the theoretical analysis**
>
> Recently, I carefully reviewed the theoretical analysis section again and discovered some important issues. I hope the author can provide a response during the discussion stage so that I can better examine the contribution of this article.
>
> 1. Formula 15 has an error, it should be $I(f(\tilde{G});Y) = H(Y)-H(Y|f(\tilde{G})$
> 2. Formula 17 in the appendix is correct, but it does not correspond to Formula 2 in Theorem 3.1 in the main text. But this is actually more reasonable, as minimizing $H(Y|f(\tilde{G}))$ is equivalent to optimizing the downstream task of OOD generalization.
> 3. The assumption in Equation 22 is quite strong. Why is $L_s$ proportional to $|G_s|$? This assumption is central to the derivation of the generalization bound in this work and requires careful consideration.
> 4. In Formula 42, why is $E[X_{ij}^{'}] = 1/|G_s|$? I believe, based on the author's alignment of the edge sampling probability for the spurious subgraph to a uniform distribution ($\mathcal{L}_{div}$), it should be 1/2.
> 5. I also have doubts about the derivation of formula 36. Based on my step-by-step deduction, formula 35 remains correct. However, the conclusion of formula 36 should be based on a probability of $1-\delta$ that $|L_c(\theta;D) - L_c(\theta; S)| \leq 2\sqrt({\frac{ln(4|\Theta|-ln\(\delta)}{2n}})$.
>
> Based on the above concerns about theoretical analysis, I have to unfortunately lower my score judgment first. I strongly encourage the theoretical contribution of this article, but enough clarify is needed. I look forward to the author's further response, so that I can revise my views on these issues.

---

> ### Author Response · Authors · 2024-11-20
> **Reply to Reviewer Ww8s (Part 1/2)**
>
> *We sincerely thank you for your valuable feedback and constructive suggestions in reviewing our paper! We also appreciate your follow-up careful comments on the theoretical aspects. Please see below for our responses to your comments and concerns.*
>
> >**W1: Why graph size constraint loss as expected?**
>
> **Responses:** Thank you for your insightful question! The graph size constraint $\mathcal{L}\_e$ alone indeed cannot guarantee that the removed edges belong solely to the spurious subgraph. However, as we demonstrate in Proposition 2, when $\mathcal{L}\_e$ acts as a regularizer of Empirical Risk Minimization (ERM) and under Assumption 2, $\mathcal{L}\_e$ will effectively prune only the spurious edges. The high-level reasoning is that when $I(G_c; Y) > I(G_s; Y)$, ERM tends to retain $G_c$ to support accurate predictions, which encourages pruning of edges in $G_s$ instead.
>
> > **W2: How to choose $K$ and $\eta$ appropriately.**
>
> **Responses:** When no prior knowledge is given, it indeed presents a challenge to choose suitable values for $K$ and $\eta$. However, through our hyperparameter sensitivity conducted on multiple real-world datasets, we observe that $K$ and $\eta$ only require searching within a limited range (e.g., $\eta \in \\{0.5, 0.75, 0.85\\}$ and $K \in \\{50, 70, 90\\}$). In addition, our hyper-parameter sensitivity analysis (Figures 2 and 6) demonstrates that our method can stably outperform the baseline methods across a variety of real-world datasets. This stability alleviates the challenge of selecting hyperparameters.
>
> > **W3: Including more related work in discussion.**
>
> **Responses:** As suggested by the reviewer, we have added more discussion on the mentioned studies in Section 5, highlighted in blue in our revised draft.
>
> > **Q1-2: Issues in Eqn.15 and Eqn.17**
>
> **Responses:** Thank you for the great catch! We have corrected Eqn.15 accordingly in line 908 in the revised PDF. In addition, we have updated Eqn.2 in line 154 to make it consisitent with Eqn.17, both are highlighted in blue.
>
> > **Q3: Why $L_s$ is proportional to $|G_s|$?**
>
> **Responses:** Thank you for your insightful comments! This assumption is motivated by, and aligns closely with the empirical observations. For instance, in the _SPMotif datasets_: $G_s$ and $Y$ are correlated with a predefined bias $b$ in training set, and the correlation strength is equal across the validation and test datasets. Additionally, there is a graph size shift in the test dataset, where $|G_s|$ is larger than that in the training and validation sets, the test performance is significantly worse than validation performance due to the increase in $|G_s|$.
>
> Similarly, for any dataset experiencing a graph size shift, we can observe the same trend. Therefore, we constrain the form of $L_s(\theta ; D)$ to be proportional to $|G_s|$, which is consistent with our experimental findings, i.e., as $|G_s|$ increases, the OOD performance gets worse.
>
> > **Q4: Derivation in Eqn.42**
>
> **Responses:** Thank you very much for catching this typo in our derivation. We have corrected the proof and provided an updated version, with changes highlighted in blue starting from line 1117. We would also like to clarify that this typo does not compromise the validity of our theoretical results. To facilitate the reviewing, we present the key part of the revised proof as follows:
>
> $\mathbb{E}[W_c] = \mathbb{E}[\sum_{e_{ij} \in G_c} X_{ij}] = \sum_{e_{ij} \in G_c} \mathbb{E}[X_{ij}] = \left|G_c\right|,$
>
> $\mathbb{E}[W\_s^{\mathcal{P}}] = \mathbb{E}[\sum_{e_{ij} \in G_s^{\mathcal{P}}} X'\_{ij}] = \sum_{e_{ij} \in G_s^{\mathcal{P}}} \mathbb{E}[X'_{ij}] = \frac{|G_s^{\mathcal{P}}|}{2} = \frac{\eta|\mathcal{E}|-|G_c|}{2}.$
>
>
> Therefore, given a suitable $\eta$ that prunes spurious edges from $G_s$, $2|G_c| \gg \eta |\mathcal{E}| -|G_c|$, i.e., $\mathbb{E}[t^*(G)]$ will be dominated by $G_c$ in terms of edge probability mass, therefore, we conclude that $G_c \approx \mathbb{E}[t^*(G)]$.
>
>
> __Discussion:__ This theoretical result also implies a trade-off between the distinguishability of $G_c$ in $t(\cdot)$ and the capability to extrapolate to new environments. Specifically, setting $p < 1/2$ in $\mathcal{L}\_{div}$ can better guide $t(G)$ to identify $G_c$, as it emphasizes the extraction of $G_c$. However, this comes at the cost of reduced diversity of the generated subgraphs, which hurts the model's OOD generalization performance in novel environments. Thus, setting $p = 1/2$ in $\mathcal{L}\_{div}$ strikes a balance between the effectiveness of identifying $G_c$ and maintaining maximal diversity for environments extrapolation.

---

> ### Author Response · Authors · 2024-11-20
> **Reply to Reviewer Ww8s (Part 2/2)**
>
> > **Q5: Derivation in Eqn.36**
>
> **Responses:** We sincerely appreciate your constructive suggestions in helping us improve the quality of our paper. Upon carefully double-checking our proof, we found that we indeed missed the coefficient 2 before $\sqrt{\frac{\ln(4|\Theta|)-\ln (\delta)}{2 n}}$ in Eqn.36. We have corrected this issue in our updated draft, and further improved the clarity of the derivation, with the changes highlighted in blue.
>
> To facilitate the reviewer's reading process, we provide the changed part of the revised proof below:
>
> To upper bound Eqn.31 with probability $1-\delta$, for both terms $\left|\mathbb{E}\left[L_c(\theta ; D)\right]-L_c(\theta ; D)\right|$ and $\left|\mathbb{E}\left[L_c(\theta ; S)\right]-L_c(\theta ; S)\right|$, we set the right-hand side of Eqn.34 and Eqn.35 as $\delta/2$, that is,
>
> $$
> 2|\Theta| \exp \left(-2 \epsilon^2 n\right)=\frac{\delta}{2} \Rightarrow  \delta=4|\Theta| \exp \left(-2 \epsilon^2 n\right).
> $$
>
> Therefore, we conclude that with probability at least $1-\delta$, we have
>
> $$
> |L_c(\theta ; D) - L_c(\theta ; S)| \leq 2\sqrt{\frac{\ln(4|\Theta|)-\ln (\delta)}{2 n}}.
> $$
>
> Finally, we get:
>
> $$
> |L(\theta ; D) - L(\theta ; S)| \leq 2(c|G_s| + 1) \sqrt{\frac{\ln(4|\Theta|)-\ln (\delta)}{2 n}}.
> $$
>
> Once again, thank you for your very careful reading, especially the appendix section - it makes for a significant improvement in the quality of our manuscript!
>
> ---
>
> We are grateful for your positive evaluation of our presentation, novelty, and effectiveness. We hope our responses address your concerns. We appreciate your support and we are happy to engage further if there are any other point we missed! Thank you!

---

> > ### Comment · Reviewer_Ww8s · 2024-11-24
> >
> > Thank you for your response to my questions. To be frank, for Q3, this assumption is too strong and is closely related to the conclusion you are trying to reach. As for Q4, the scaling relationship is not that elegant. Overall, however, the authors have made a good attempt for the theoretical analysis of augmentation-based graph OOD problem.

---

> ### Author Response · Authors · 2024-11-25
> **Addressing the concerns on theoretical aspects**
>
> Dear Reviewer Ww8s, thank you for your follow-up comment. We wish to address your concerns as follows:
>
> #### **Assumption of $L_s(\theta; \cdot)$ Proportional to $|G_s|$**
>    - In addition to the motivations mentioned in our previous discussions, we clarify that $L_s(\theta; \cdot)$ is defined as a summation over all spurious edges in $G_s$ (as shown in Eq. 21). Therefore, we respectfully argue that assuming proportionality to $|G_s|$ is a reasonable and rational assumption.
>    - While the definition $L_s(\theta ; D):= c\left|G_s\right| L_c(\theta ; D)$ may seem restrictive, it maintains sufficient flexibility since $c$ can absorb other relevant factors such as the model architecture and the topological complexity of the graph. This form ensures adaptability across different datasets and models.
>    - Furthermore, this OOD generalization bound, inspired by Assumption 1, accurately reflects the OOD generalization performance across datasets. For example, in the Motif-base dataset, where $|G_s|$ is smaller, all methods achieve better OOD generalization performance. Conversely, in the Motif-size dataset, where $|G_s|$ constitutes a significant portion of the graph, the bound becomes looser, resulting in worse OOD performance across all methods.  This demonstrates the broad applicability and validity of the proposed generalization bound.
>
> #### **Scaling Factor in Theorem 4.2**
>    - We thank the reviewer’s feedback again, which allowed us to refine the scaling factor in our revision.
>    - The scaling factor is directly associated with the diversification strength. The factor of $2$ represents a trade-off between improved extrapolation capability and explainability, balancing these two objectives.
>    - This theoretical result implies that, under in-distribution settings, we can set the aligning probability to be less than $0.5$ to prioritize explainability while de-emphasizing environment extrapolation. This highlights the flexibility of our proposed framework, iSSD, to adapt to different use cases.
>
> We hope our response resolves the reviewer’s concerns about the theoretical aspects of our work. Please do not hesitate to reach out if you have further questions. Thank you for your valuable comments!

---

> ### Author Response · Authors · 2024-12-02
> **Gentle reminder**
>
> Once again, we are grateful for your time and effort for reviewing our paper. Since the discussion period will end in a few days, we are very eager to get your feedback on our response. We understand that you are very busy, but we would highly appreciate it if you could take into account our response and reconsider your evaluation if some concerns are addressed. Thank you!
>
> Authors

---

### Official Review · Reviewer_tUpV · 2024-11-03

**Soundness:** 3
**Presentation:** 3
**Contribution:** 2
**Rating:** 5
**Confidence:** 3

**Summary:**

One big challenge for graph machine learning (GML) is that real-world graph data continuously evolves over time, introducing changes in graph structure and node/edge features, causing graph distribution shift. However, retaining GNN models every time the graph is updated is expensive or sometimes infeasible. How to handle the out-of-distribution (OOD) problem in GNN model training becomes a challenging problem. This paper proposes a novel theory to identify the invariant subgraphs, whose edges exhibits high predicted probabilities in the learnable data transformation to the target graph labels, and the spurious subgraph, whose edges exhibit lowest predicted probabilities. The proposed  learning framework based on the theory exhibits stable and good performance over existing baseline models on 7 real-world datasets with an average of 2.38%

**Strengths:**

* The paper proposes a novel theory to identify the invariant subgraphs and spurious subgraphs. The paper shows that with the proposed edge dropping function t, the graph size constraint loss L_e and the spurious subgraph diversification loss L_{div}, the proposed method can identify the invariant graphs and the spurious graphs. Furthermore, the evaluation in section 7.5 demonstrates that the proposed method can distinguish G_c and G_s using the GOOD-Motif datasets.
* The proposed methods shows stable and good performance over existing baseline models on 7 real-world datasets with an average of 2.38%.

**Weaknesses:**

* The whole method assumes that for every label y, there exists only one G_c. Is it possible that there are more than one G_c correspond to a label y? Will all the assumptions and theorems hold in such cases?
* The proposed method can not handle OOD cases on graph tasks like node classification, link prediction, etc. This limit the scope of the method.
* Hyper-parameter \eta (L215) and K (L249) are critical to the size of G_s and are thus critical to the method. Theorem 4.1 and 4.2 rely on these two hyper-parameters. How to select the right value for these hyper-parameters without knowing the size of G_c is not discussed.
* Some proves are missing:
    * Why Theorem 3.1 is hold is not proved.
    * Why P(X^’_{ij}) = 1/|G_s| is enforced by L_{div} is not well explained.

**Questions:**

I would suggest to move definition 2 before Theorem 3.1.

---

> ### Author Response · Authors · 2024-11-20
> **Reply to Reviewer tUpV (Part 1/3)**
>
> *We sincerely thank you for your valuable feedback and constructive suggestions in reviewing our paper! We are encouraged that you acknowledged we propose a novel theory to identify the invariant subgraphs and spurious subgraphs with comprehensive experiments. Please see below for our responses to your comments and concerns.*
>
> ---
>
> >**W1: Assumption of multiple $G_c$**
>
> **Responses:** Thank you for your insightful question! We agree that in real-world graph applications, it is indeed possible for a graph to contain multiple invariant substructures that are causally related to $Y$.
>
> Despite assuming a single $G_c$ as in most of the previous studies [1-3], we emphasize our approach can be naturally extended to the case of multiple $G_c$.
>
> - Specifically, suppose there are $K$ invariant subgraphs, denoted as $G_{c,i}$ for $i \in [K]$. For any specific $G_{c,i}$, we can redefine the spurious subgraph $G_s'$ as $G_s' = G_s \cup \\{ G_{c,j} \mid j \neq i \\}$. Under the presence of $G_s$, our assumption $I(G_{c,i}; Y) > I(G_s'; Y)$ remains valid for any $i \in [K]$. Therefore, the assumptions and theoretical results in our work remain valid even when multiple $G_c$ exist in the datasets.
> - To verify the effectiveness of our method in the presence of multiple $G_c$, we added experiments on a curated dataset based on SPMotif. In the train/valid/test datasets, two invariant substructures are attached to the spurious subgraph. The results are shown below.
>
>
> Table: Experimental results on the SPMotif datasets with 2 invariant subgraphs.
> | Method  | $b=0.40$             | $b=0.60$             | $b=0.90$             |
> |---------|------------------------------------------|------------------------------------------|------------------------------------------|
> | ERM     | 53.48±3.31                               | 52.59±4.61                               | 56.76±8.06                               |
> | IRM     | 52.47±3.63                               | 55.62±7.90                               | 48.66±2.33                               |
> | VRex    | 49.68±8.66                               | 48.89±4.79                               | 47.97±2.61                               |
> | GSAT    | 59.34±7.96                               | 58.43±10.64                              | 55.68±3.18                               |
> | GREA    | 64.87±5.76                               | 67.66±6.29                               | 59.40±10.26                              |
> | CIGA    | 69.74±6.81                               | 71.19±2.46                               | *65.83±10.41*                            |
> | AIA     | **71.61±2.09**                           | *72.01±2.13*                             | 58.14±4.21                               |
> | iSSD (Ours)    | *70.41±7.53*                             | **74.61±3.17**                           | **66.75±4.33**                           |
>
> The above results validate the effectiveness of our approach given the presence of multiple $G_c$.
>
> We have added a detailed discussion and new experimental results on the multiple $G_c$ scenarios in Appendix H.3 in the revised draft, highlighted in blue.

---

> > ### Comment · Reviewer_tUpV · 2024-11-20
> > **Response**
> >
> > I like the discussion of multiple $G_c$ here.
> > But can you be more specific on what are the two invariant substructures you attached to spurious subgraphs on SPMotif?

---

> ### Author Response · Authors · 2024-11-20
> **Reply to Reviewer tUpV (Part 2/3)**
>
> > **W2: Only applicable for graph-level OOD generalization**
>
>
> **Responses:**  Thank you for the insightful comment. We would like to first clarify the unique challenges and inherent differences between node-level (link-level) and graph-level OOD generalization, then discuss the potential of extending our method to tackle the node-level OOD generalization.
>
> - Topological characteristics: Graph-level tasks treat an entire graph as a single instance, whereas node-level tasks take the entire graph as input and focus on individual nodes. As a result, node-level OOD methods often exploit topological information from the graph structure [4-5], while such information is not applicable for graph-level tasks.
> - Dataset characteristics: Many graph-level OOD datasets, particularly in the molecular domain, have very limited or even no node features, relying heavily on graph structures. In contrast, node-level OOD datasets are typically rich in node features, which presents distinct challenges and opportunities.
> - Different types of distribution shifts: The types of distribution shifts also differ between the two levels. For example, molecular datasets commonly exhibit scaffold shifts and size shifts, which are prevalent in graph-level tasks. In contrast, temporal shifts and domain shifts are more frequently observed in node-level OOD datasets.
>
> Given these distinctions, methods for handling graph-level and node-level (or link-level) OOD generalization have been developed in parallel to address their respective unique challenges. As such, in this work, we focus on comparisons with graph-level invariant learning and data augmentation methods that align with the scope of our work.
>
> Despite the different challenges, we find it is beneficial to discuss the potential for our method to adapt to node-level and link-level tasks. For instance, our approach could be extended by diversifying the $K$-hop subgraphs for each node to weaken spurious correlations. Considering the higher information density in node features for node-level and link-level tasks, we could further diversify node features as part of the augmentation strategy as an enhancement. Similarly, for node-level and link-level tasks, we could constrain the size of ego-networks and the information density of node features to improve generalization. These extensions present promising directions for our future work.
>
>
> To enhance the applicability of our approach, we have added a discussion on the applicability of iSSD for node-level and link-level OOD generalization in Appendix C.
>
> > **W3: Hyper-parameter selection.**
>
> **Responses:**  Thank you for the great question! In practice, the selection of these hyper-parameters can be approached in two ways depending on the availability of domain knowledge:
>
> - With domain knowledge: When domain knowledge is available, e.g., we know the percentage of the spurious edges among all edges, $\eta$ and $K$ can be directly set to suitable values without the need for extensive tuning.
> - Without domain knowledge: We agree with the reviewer that choosing appropriate values for $\eta$ and $K$ in the absence of domain knowledge could be challenging. However, our hyper-parameter sensitivity analysis (Figures 2 and 6) demonstrates that our method can stably outperform the baseline methods across different hyperparameter settings on a variety of datasets. This stability alleviates the challenge of selecting hyperparameters.

---

> > ### Comment · Reviewer_tUpV · 2024-11-20
> > **Response to graph-level OOD generalization**
> >
> > Can you provide more evidence that the proposed method can be used for node-level or link-level OOD cases? In node-level or link-level OOD cases, the local topology of certain nodes may change when new nodes and edges are added into a graph. Can identifying invariant subgraphs solve such a problem?

---

> ### Author Response · Authors · 2024-11-20
> **Reply to Reviewer tUpV (Part 3/3)**
>
> > **W4: Minor issues in proofs.**
>
> **Responses:** Please see our response below.
>
> **The proof of Theorem 3.1**
>
> We wish to clarify that the proof of Theorem 3.1 can be found in Appendix E.1. To improve clarity, we have added: "We prove Theorem 3.1 in Appendix E.1" in line 157, and highlighted in blue.
>
> Why $P(X_{ij}^{'})=1/|G_s|$
>
> We appreciate the reviewer's observation regarding the issues in our proof. Specifically, $P(X_{ij}^{'})$ should be $1/2$. We have revised the relevant sections of the proof accordingly and highlighted the changes in blue, from lines 1117 to 1120. To facilitate the reviewer's reading, we provide the changed part of the revised proof below:
>
>
> $\mathbb{E}[W_c] = \mathbb{E}[\sum_{e_{ij} \in G_c} X_{ij}] = \sum_{e_{ij} \in G_c} \mathbb{E}[X_{ij}] = \left|G_c\right|,$
>
>
> $\mathbb{E}[W_s^{\mathcal{P}}] = \mathbb{E}[\sum_{e_{ij} \in G_s^{\mathcal{P}}} X'\_{ij}] = \sum_{e_{ij} \in G_s^{\mathcal{P}}} \mathbb{E}[X'\_{ij}] = \frac{|G_s^{\mathcal{P}}|}{2} = \frac{\eta|\mathcal{E}|-|G_c|}{2}.$
>
> Therefore, given a suitable $\eta$ that prunes spurious edges from $G_s$, $2|G_c| \gg \eta |\mathcal{E}| -|G_c|$, i.e., $\mathbb{E}[t^*(G)]$ will be dominated by $G_c$ in terms of edge probability mass, therefore, we conclude that $G_c \approx \mathbb{E}[t^*(G)]$.
>
>
> **Discussion.** This theoretical result also implies a trade off between the distinguishability of $G_c$ in $t(\cdot)$ and the capability for environment extrapolation. Specifically, setting $p < 1/2$ in $\mathcal{L}\_{div}$ can better guide $t(G)$ to identify $G_c$, as it emphasizes the extraction of $G_c$. However, this comes at the cost of reduced diversity in the generated subgraphs, which hurt the model's OOD generalization performance in novel environments. Thus, setting $p = 1/2$ in $\mathcal{L}\_{div}$ strikes a balance between the effectiveness of identifying $G_c$ and maintaining sufficient diversity for environment extrapolation.
>
>
> > **Q1: Paper structure.**
>
> **Responses:**  Thank you for the suggestion! We found your suggestion useful and have placed Def.2 before Theorem 3.1.
>
>
> ---
>
> We sincerely thank you for your insightful questions and approval of the presentation, novelty, and effectiveness. We hope we have addressed your concerns.
>
> ---
>
> __References__
>
>
> [1] Wu, et al., Discovering invariant rationales for graph neural networks, ICLR 2022.
>
> [2] Chen, et al., Learning causally invariant representations for out-of-distribution generalization on graphs, NeurIPS 2022.
>
> [3] Li, et al., Learning invariant graph representations for outof-distribution generalization, NeurIPS 2022.
>
> [4] Wu, et al., Handling Distribution Shifts on Graphs: An Invariance Perspective, ICLR2022.
>
> [5] Wang, et al., Dissecting the Failure of Invariant Learning on Graphs, NeurIPS2024.

---

> ### Author Response · Authors · 2024-11-21
> **More discussions on multple $G_c$ assumption**
>
> Thank you for recognizing our discussion of the multiple $G_c$ assumption. Please see our response below for more details on the curated SPMotif datasets.
>
> *More Details on SPMotif Datasets with Two $G_c$:*
>
> Vanilla SPMotif Datasets:  In the vanilla SPMotif datasets, each graph is generated as follows:
>   1. A spurious subgraph is first generated from a set of predefined base graphs (Tree, Ladder, Wheel).
>   2. An invariant subgraph (House, Cycle, Crane) is then randomly selected with equal probability and attached to the spurious subgraph.
>
> Modified SPMotif Datasets with Two $G_c$:  To align with the multiple $G_c$ assumption, we modify the dataset generation process as follows:
>   1. A spurious subgraph is first generated from the same predefined base graphs (Tree, Ladder, Wheel).
>   2. One invariant subgraph pattern (House, Cycle, Crane) is randomly selected with equal probability, and 2 subgraphs of the selected pattern (e.g., Cycle) are randomly attached to the spurious subgraph at different locations.
>
> By introducing two invariant subgraphs per graph, we can evaluate the performance of various OOD methods under the new assumption of multiple $G_c$.
>
> ---
>
> We hope this clarifies the details of the SPMotif dataset generation and how it accommodates the multiple $G_c$ assumption. Please feel free to reach out if any further questions!

---

> ### Author Response · Authors · 2024-11-21
> **More discussions on the applicability for node-level and link-level OOD tasks**
>
> Thank you for the follow-up and inspiring question. Please see our response below for *more evidence on why iSSD can be extended to node-level and link-level OOD cases:*
>
> - **Similar Assumptions:**  Previous studies on node-level OOD problems often share similar assumptions to our work, as discussed in [1, 2, 3]. A key distinction is that, instead of assuming an invariant subgraph, they generally assume a $K$-hop ego-network $G_i$ (with invariant features) for each node $v_i$, which is both stable and sufficiently predictive of the target label $y$. To enhance OOD generalization performance, these methods also aim to identify invariant substructures (and node features). This shared assumption provides a solid foundation for extending iSSD to node-level and link-level OOD problems.
>
>
> - **Connections to Existing Work:**   Previous studies, such as EERM [1], also employ multiple context generators to create subgraphs by maximizing variance loss, which conceptually aligns with the idea of diversifying subgraphs. In contrast, we adopt a different approach for maximally diversifying subgraphs. This similarity supports our claim that our method holds great potential for node-level and link-level OOD tasks.
>
>
> - **Addressing New Node or Link Emergence:**  We agree with the reviewer that the emergence of new nodes or links poses additional challenges for node-level OOD. However, this inductive setting closely resembles the graph-level OOD setting, as graph-level datasets are inherently inductive (each test sample is unseen), unlike the transductive setting in traditional node-level datasets. Below, we discuss why iSSD can handle emergent nodes and links:
>
>   - New links emerge: By diversifying edges with the lowest probabilities, and assuming that for each node $v_i$ an invariant substructure exists, our approach can diversify spurious edges similarly to the graph-level OOD setting. When new links emerge, our method remains effective in identifying critical edges, as the diversification process extrapolates to many unseen environments. This allows the model to generalize better in link-level OOD tasks.
>   - New nodes emerge: Similarly, iSSD can effectively handle cases where new nodes emerge in the ego-network. By extrapolating to unseen environments, nodes that are not critical to the central node $v_i$ are likely to be excluded. For classifying new nodes, the presence of rich node features, and given that *similar nodes tend to share similar invariant structures*, allows our method to adapt effectively to unseen nodes by leveraging similarities with nodes in the training set.
>
>
> In summary, while challenges remain, the shared assumptions and superior capability of iSSD to extrapolate to unseen environments imply that iSSD has significant potential to be adapted for node-level and link-level OOD tasks.
>
> ---
>
> Please kindly let us know if our explanations above clarify your concerns. Otherwise, we would be happy to have a further discussion!
>
> ---
>
> **References**
>
> [1] Wu, et al., Handling Distribution Shifts on Graphs: An Invariance Perspective, ICLR2022.
>
> [2] Gui, et al., GOOD: A Graph Out-of-Distribution Benchmark, NeurIPS2022
>
> [3] Wang, et al., Dissecting the Failure of Invariant Learning on Graphs, NeurIPS2024

---

> > ### Comment · Reviewer_tUpV · 2024-12-02
> > **Response**
> >
> > I understand that the proposed method has potential for adaptation to node-level and link-level OOD tasks. However, there is proof to demonstrate its effectiveness (Theoretical proof and evaluations).

---

> ### Author Response · Authors · 2024-11-25
>
> Dear Reviewer tUpV,
>
> As the discussion deadline approaches, we are wondering whether our responses have properly addressed your concerns and follow-up questions? Your feedback would be extremely helpful to us. If you have further comments or questions, we hope for the opportunity to respond to them.
>
> Many thanks,
>
> Authors

---

> ### Author Response · Authors · 2024-12-02
> **Adaptation to node-level OOD (Part 1/3)**
>
> Thank you for your response! To address your concern, we have provided the following discussions on how to adapt iSSD to node-level OOD problem, and provide a proof for it. We begin by stating the assumptions, followed by the adaptation and proof.
>
> ### Assumptions for Node-Level OOD Generalization
>
> **Assumption 1:**
> Given a graph $G$, consider the $l$-layer ego-network of a node $v_i$, denoted as $G^{(i,l)}$. Within this ego-network, there exists an invariant subgraph $G_c^{(i,l)}$ and invariant features $H_c^{(i,l)}$ stable across environments $e$ and $e'$, satisfying:
> 1. For any environments $e, e' \in \mathcal{E}_{tr}$,
>    $$
>    P^e(Y \mid G_c^{(i,l)}) = P^{e'}(Y \mid G_c^{(i,l)}) \quad \text{and} \quad P^e(Y \mid H_c^{(i,l)}) = P^{e'}(Y \mid H_c^{(i,l)}).
>    $$
>
> 2. The target $Y$ can be expressed as
>    $$
>    Y = f^*(G_c^{(i,l)}, H_c^{(i,l)}) + \epsilon,
>    $$
>    where $\epsilon \perp (G^{(i,l)}, H^{(i,l)})$ represents random noise, and $\perp$ indicates statistical independence.
>
> **Assumption 2:**
> For the $l$-layer ego-network of a node $v_i$, the mutual information between the invariant components and the target $Y$ satisfies:
> 1. For the subgraph, we have:
>    $$
>    I(G_c^{(i,l)}; Y) > I(G_s^{(i,l)}; Y).
>    $$
> 2. For the node-level features, we have:
>    $$
>    I(H_c^{(i,l)}; Y) > I(H_s^{(i,l)}; Y).
>    $$
>
> These assumptions ensure that the invariant subgraph and features are both more stable and more predictive of $Y$ compared to their spurious counterparts. With the above extended assumptions for node-level or link-level OOD settings, the size constraint can be extended further to account for node features:
>
> $$
> \mathcal{L}\_e = \mathbf{E}_\{\mathcal{G}^{(i,l)}}\left[\left(\frac{\sum\_{(u, v) \in \mathcal{E}^{(i,l)}} \tilde{\mathbf{A}}\_{u v}^{(i)}}{|\mathcal{E}^{(i,l)}|} - \eta\right)^2 + \lambda \|H^{(i,l)}\|_1\right],
> $$
>
> where:
> - $\mathcal{E}^{(i,l)}$ is the edge set of the $l$-layer ego-network for node $v_i$,
> - $\tilde{\mathbf{A}}_{uv}^{(i)}$ is the adjacency matrix after applying $t(\cdot)$,
> - $\|H^{(i,l)}\|_1$ represents the L1 norm of the node features in the $l$-layer ego-network, added to induce sparsity,
> - $\eta$ is a hyperparameter controlling the edge budget, and $\lambda$ balances the graph size and feature regularization terms.
>
> This constraint encourages sparsity in both the structure and features of the ego-network, ensuring that spurious components are pruned while invariant features are preserved.
>
> Next we prove that: Under Assumption 2, the size and feature constraint loss $\mathcal{L}\_e$, when acting as a regularizer for $\mathcal{L}\_{GT}$, will prune edges from the spurious subgraph $G\_s^{(i,l)}$ and features from $H_s^{(i,l)}$, while preserving $G_c^{(i,l)}$ and $H_c^{(i,l)}$.

---

> ### Author Response · Authors · 2024-12-02
> **Adaptation to node-level OOD (Part 2/3)**
>
> **Proof (Graph Structure):**
> We start by expanding the cross-entropy loss $\mathcal{L}\_{GT}$:
>
> $$
> \mathcal{L}\_{GT} = -\mathbb{E}\_{G^{(i,l)}}[\log \mathbb{P}(Y \mid f(\tilde{G}^{(i,l)}))],
> $$
>
> where $\tilde{G}^{(i,l)} \sim t(G^{(i,l)})$. Assume that $|\tilde{G}^{(i,l)}| > |G_c^{(i,l)}|$, which is controlled by the hyperparameter $\eta$. Further, assume that $\tilde{G}^{(i,l)}$ does not include the invariant subgraph $G_c^{(i,l)}$.
>
> Let a subgraph $g$ be subtracted from $\tilde{G}^{(i,l)}$ such that $|g| = |G_c^{(i,l)}|$. Define:
>
> $$
> G' = \tilde{G}^{(i,l)} \setminus g,
> $$
>
> and construct a new graph:
>
> $$
> G' \cup G_c^{(i,l)}.
> $$
>
> By Assumption 1, the invariant subgraph $G_c^{(i,l)}$ holds sufficient predictive power for $Y$, and by Assumption 2, $G_c^{(i,l)}$ is more informative to $Y$ than $G_s^{(i,l)}$. Thus:
>
> $$
> \mathbb{P}(Y \mid f(G' \cup G_c^{(i,l)})) > \mathbb{P}(Y \mid f(G' \cup g)), \quad \forall g \subseteq \tilde{G}^{(i,l)}.
> $$
>
> As a result, $\mathcal{L}_{GT}$ will decrease when $G_c^{(i,l)}$ is included in $\tilde{G}^{(i,l)}$. Therefore, the graph size regularization term $\mathcal{L}_e$ ensures that $\tilde{G}^{(i,l)}$ contains $G_c^{(i,l)}$ and prunes edges from $G_s^{(i,l)}$.
>
> **Proof (Node Features):**
> Suppose the optimal feature set $\tilde{H}^{(i,l)}$ is encoded by a GNN encoder $h(\cdot)$ from the $l$-layer ego-network $G^{(i,l)}$ and does not include the invariant features $H_c^{(i,l)}$. Define a feature set:
>
> $$
> H' = \tilde{H}^{(i,l)} \setminus h,
> $$
>
> where $|h| = |H_c^{(i,l)}|$, and construct:
>
> $$
> H' \cup H_c^{(i,l)}.
> $$
>
> Under Assumption 1, the invariant features $H_c^{(i,l)}$ retain sufficient predictive power for $Y$, and by Assumption 2 for features:
>
> $$
> I(H_c^{(i,l)}; Y) > I(H_s^{(i,l)}; Y).
> $$
>
> Thus, adding $H_c^{(i,l)}$ to $\tilde{H}^{(i,l)}$ will increase the certainty of predictions:
>
> $$
> \mathbb{P}(Y \mid f(G^{(i,l)}, H' \cup H_c^{(i,l)})) > \mathbb{P}(Y \mid f(G^{(i,l)}, H' \cup h)), \quad \forall h \subseteq \tilde{H}^{(i,l)}.
> $$
>
> However, by definition, $\tilde{H}^{(i,l)}$ is the feature set output by the encoder $h(\cdot)$ acting on $G^{(i,l)}$. If $H_c^{(i,l)}$ were excluded, the encoded features $\tilde{H}^{(i,l)}$ would lead to  reduced predictive signals for $Y$ under Assumption 2. This leads to a contradiction, as including $H_c^{(i,l)}$ would always produce a lower loss $\mathcal{L}_{GT}$, making it part of the optimal solution.
>
> ### Extended Formulation for $\mathcal{L}_{\text{div}}$
>
> To incorporate diversification for both spurious subgraphs and node features, we extend the regularization term $\mathcal{L}\_{div}$ as follows:
>
> $$
> \mathcal{L}\_{\text{div}} = \mathbb{E}\_{\mathcal{G^{(i,l)}}} \frac{1}{|\mathcal{E}\_s^{(i,l)}|} \sum_{e\_{uv} \in \mathcal{E}\_s^{(i,l)}} \left| w_{uv} - \frac{1}{|\mathcal{E}\_s^{(i,l)}|} \right| + \lambda \cdot \mathbb{E}\_{\mathcal{G^{(i,l)}}} \mathrm{KL}(\tilde{H}^{(i,l)}, \mathcal{U}),
> $$
>
> where:
> - $\mathcal{E}\_s^{(i,l)}$ represents the set of spurious edges in the $l$-layer ego-network of node $v_i$,
> - $w_{uv}$ is the normalized probability for edge $e_{uv}$,
> - $\mathrm{KL}(\tilde{H}^{(i,l)}, \mathcal{U})$ is the KL-divergence between the encoded node features $\tilde{H}^{(i,l)}$ and a uniform distribution $\mathcal{U}$,
> - $\lambda$ balances the contributions of graph and feature diversification.
>
> The first term diversifies the spurious subgraph, while the second term encourages the node features $\tilde{H}^{(i,l)}$ to align with a uniform distribution, increasing uncertainty in the spurious features.
>
> ---
> Next we prove that the KL term will diversify the spurious node features by aligning it with uniform distribution.
>
> ### Proof for the Extended $\mathcal{L}\_{\text{div}}$
>
> **Objective:** Maximize the mutual information (MI):
> $$
> I(\tilde{H}^{(i,l)}; Y) + \eta I(\tilde{H}^{(i,l)}; \mathcal{U}),
> $$
> where $\eta$ is a small constant (e.g., 0.01) that ensures feature alignment with the uniform distribution does not dominate predictive accuracy.
>
> ---
>
> #### Step 1: Expand $\tilde{H}^{(i,l)}$ into invariant and spurious components
>
> We decompose $\tilde{H}^{(i,l)}$ as:
> $$
> \tilde{H}^{(i,l)} = H_c^{(i,l)} \cup H_s^{(i,l)},
> $$
> where $H_c^{(i,l)}$ and $H_s^{(i,l)}$ denote the invariant and spurious features, respectively.
>
> Using the chain rule for MI:
> $$
> I(\tilde{H}^{(i,l)}; Y) = I(H_c^{(i,l)}; Y) + I(H_s^{(i,l)}; Y \mid H_c^{(i,l)}),
> $$
> $$
> I(\tilde{H}^{(i,l)}; \mathcal{U}) = I(H_c^{(i,l)}; \mathcal{U}) + I(H_s^{(i,l)}; \mathcal{U} \mid H_c^{(i,l)}).
> $$

---

> ### Author Response · Authors · 2024-12-02
> **Adaptation to node-level OOD (Part 3/3)**
>
> #### Step 2: Simplify Using Assumptions
>
> 1. **For $I(\tilde{H}^{(i,l)}; Y)$:**
>    - By Assumption 2, $H_c^{(i,l)}$ is highly predictive of $Y$, while $H_s^{(i,l)}$ is not.
>    - Thus, $I(H_s^{(i,l)}; Y \mid H_c^{(i,l)}) = 0$, leaving:
>      $$
>      I(\tilde{H}^{(i,l)}; Y) = I(H_c^{(i,l)}; Y).
>      $$
>
> 2. **For $I(\tilde{H}^{(i,l)}; \mathcal{U})$:**
>    - Due to the regularization effect of empirical risk minimization (ERM), $H_c^{(i,l)}$ is aligned with the target $Y$ and does not resemble a uniform distribution, The KL-divergence term $\mathrm{KL}(\tilde{H}^{(i,l)}, \mathcal{U})$ encourages $H_s^{(i,l)}$ to align with the uniform distribution, maximizing $I(H_s^{(i,l)}; \mathcal{U} \mid H_c^{(i,l)})$. Thus, $I(\tilde{H}^{(i,l)}; \mathcal{U})$ is maximized due to the diversification of spurious node features $H_s^{(i,l)}$.
>
> ---
>
> #### Step 3: Overall Objective
>
> Substituting into the objective:
> $$
> I(\tilde{H}^{(i,l)}; Y) + \eta I(\tilde{H}^{(i,l)}; \mathcal{U}),
> $$
> we have:
> $$
> I(\tilde{H}^{(i,l)}; Y) + \eta I(\tilde{H}^{(i,l)}; \mathcal{U}) = I(H_c^{(i,l)}; Y) + \eta I(H_s^{(i,l)}; \mathcal{U} \mid H_c^{(i,l)}).
> $$
>
> - The first term, $I(H_c^{(i,l)}; Y)$, ensures that the invariant features $H_c^{(i,l)}$ remain predictive for $Y$.
> - The second term, $\eta I(H_s^{(i,l)}; \mathcal{U} \mid H_c^{(i,l)})$, guides the spurious features $H_s^{(i,l)}$ to align with the uniform distribution, reducing their influence on $Y$.
>
> ---
>
> The adaptation to the node-level OOD problem shares a conceptual similarity with the graph-level OOD problem.
> - In addition to diversifying spurious subgraphs, we propose two additional terms to enforce node-feature diversification and information constraints.
> - Furthermore, we provide a theoretical justification for the proposed regularization on node-feature constraints. However, adapting our approach to the node-level task involves significant modifications to the workload, including the implementation and benchmarking of multiple state-of-the-art methods specifically for node-level OOD tasks across various datasets, which is beyond the current scope of this work.
> - Nonetheless, the proposed adaptation is conceptually straightforward to implement and comes with theoretical guarantees.
> - It is worth emphasizing that our work aligns with prior studies, which evaluate methods under graph-level OOD setting [1-5]. By benchmarking against 17 baseline methods, we believe we have sufficiently demonstrated the superiority of iSSD in addressing graph OOD challenges.
>
> ---
>
> We hope this response addresses the reviewer’s concerns regarding the adaptation to node-level OOD problems and reconsiders  the evaluation of our work.
>
>
> ---
>
> __References__
>
> [1] Wu, et al., Discovering invariant rationales for graph neural networks, ICLR 2022.
>
> [2] Chen, et al., Learning causally invariant representations for out-of-distribution generalization on graphs, NeurIPS 2022.
>
> [3]  Liu, et al., Graph Rationalization with Environment-based Augmentations, KDD 2022
>
> [4] Sui, et al., Unleashing the Power of Graph Data Augmentation on Covariate Distribution Shift. NeurIPS 2023.
>
> [5] Gui, et al., Joint Learning of Label and Environment Causal Independence for Graph Out-of-Distribution Generalization. NeurIPS 2023.

---

### Author Response · Authors · 2024-11-20
**General Response**

Dear reviewers and AC,

We sincerely thank all reviewers and AC for their valuable feedbacks and constructive comments on our manuscript. During the rebuttal period, we have been focusing on these beneficial suggestions from the reviewers and doing our best to add several experiments and revise our draft. We believe our current carefully revised manuscript can address all the reviewers’ concerns.

As reviewers highlighted, we believe our paper tackles an important problem (**Reviewer Jzm8**) with interest for broad graph community (**Reviewer CgQ7**), introducing a novel and effective method (**Reviewer tUpV, Reviewer Ww8s, Reviewer Jzm8**) with solid theoretical foundations (**Reviewer Ww8s, Reviewer Jzm8**).


We also appreciate that the reviewers found our draft is well-written (**Reviewer Ww8s, Reviewer CgQ7**), all design choices in our method are well-motivated with sufficient empirical analysis for justification (**Reviewer CgQ7**), and detailed ablation studies and sensitivity analyses are provided (**Reviewer Jzm8**).


Moreover, we thank the reviewers for pointing out the concerns regarding the assumption and broad applicability (**Reviewer tUpV**), proof issues in the theorems (**Reviewer tUpV, Reviewer Ww8s**), the potential challenges in selecting $\eta$ and $K$ (**Reviewer tUpV, Reviewer Ww8s**), diversity of the datasets (**Reviewer CgQ7**),  the soundness and technical novelty of our proposed method (**Reviewer Jzm8**). In response to these comments, we have carefully revised and enhanced our manuscript with the following important changes with the added experiments:

- **[Reviewer tUpV]** We have added a discussion on the applicability of Assumption 1 in scenarios with multiple invariant subgraphs, supported by new empirical results for validation (in Appendix H.3).
- **[Reviewer tUpV]** We have included a discussion on extending our method to node-level and link-level OOD generalization (in Appendix C).
- **[Reviewer tUpV, Ww8s]** We have corrected issues related to Theorems 4.1 and 4.2 (in Appendix E).
- **[Reviewer Ww8s]** We have expanded Section 5 to include additional discussions on specified related works.
- **[Reviewer CgQ7]** We have conducted additional experiments on application domains beyond molecular datasets (in Appendix H.3) and added new results from an ablation study on ERM pretraining (in Appendix H.3). We have also included new experimental results addressing concept shift (in Appendix H.3).
- **[Reviewer Jzm8]** We have fixed the typo in line 72 and provided new experimental results on Recall@K and Precision@K to demonstrate the effectiveness of our method in identifying spurious edges and preserve causal edges (in Appendix H.3).

These updates are temporarily highlighted in $\textcolor{blue}{\text{blue}}$ for facilitating checking.

We hope our response and revisions address all the reviewers' concerns, and we are more than eager to have further discussions with the reviewers regarding these updates.


Thanks,

Authors

---

> ### Author Response · Authors · 2024-11-26
>
> Dear Reviewers and Area Chair,
>
> During the rebuttal phase, We identified an typo in the manuscript from our actual code implementation. Specifically, the aligning probability (in Eqn. 10) in our final implementation is not $1/2$ as stated, but rather $1/|\mathcal{E}_s|$.
>
> The uniform distribution applies to the entire estimated spurious subgraph, not to each individual edge. We have updated the draft to reflect this correction in Section 4 (line 255-263).
>
> Additionally, the proof of Theorem 4.2 have been updated accordingly (line 1119-1124). Specifically, the scaling factor is now $|\mathcal{E}_s|$ instead of 2, which provides a stronger results for: $G_c \approx \mathbb{E}_G\left[t^*(G)\right]$.
>
> Our code implementation is available at:
> [https://anonymous.4open.science/r/TempProj-1B2A](https://anonymous.4open.science/r/TempProj-1B2A)
> to facilitate code checking. The code implementation for $\mathcal{L}_{div}$ can be found in line 528 of ``modelNew/utils.py``.
>
> These updates are also highlighted in $\textcolor{blue}{\text{blue}}$ in the revised draft.
>
> We apologize for the oversight and the late update to our draft due to this inconsistency. Thank you for your understanding and for reviewing our work.

---

### Meta-Review · Area_Chair_Wd6G · 2024-12-19

**Metareview:**

This paper proposes a new environment augmentation method for graph out-of-distribution (OOD) generalization using subgraphs. The authors have made well-motivated design choices, and the empirical results provided are sufficient to demonstrate the method's potential. However, several reviewers raised important questions that warrant further discussion.

One of the primary concerns is the reliance of the method's effectiveness on certain assumptions being true. If these assumptions do not hold in real-world scenarios, the method's performance may be compromised. This concern is exacerbated by the novelty of the approach being seen as incremental rather than groundbreaking. While the authors argue that their "ranking-based approach" is novel, it is viewed by some reviewers as a relatively minor advancement over existing work. Furthermore, the soundness and generalizability of the method have been questioned by multiple reviewers. For instance, Reviewer Jzm8 raised a valid concern about the accuracy of identifying edges from spurious subgraphs, a concern that this meta-reviewer also shares.

Reviewer tUpV pointed out an important issue regarding the method's ability to handle cases with multiple invariant subgraphs for each label. The rebuttal provided additional theoretical analysis but relies on strong assumptions that could be problematic if violated. Specifically, Assumption 2 seems overly optimistic, as spurious inputs are often highly predictive of the target on the training environment, contradicting the assumption that causal subgraph signals are more predictive. Reviewer CgQ7 expressed concerns about the method's ability to identify spurious edges, the impact of concept shift, and the restrictiveness of the graph size constraint. Unfortunately, the authors' responses to these concerns were found to be somewhat unsatisfactory, as the problem seems manageable under their assumptions but may not be robust if these assumptions are only "approximately true."

On a positive note, Reviewer Ww8s had a favorable view of the paper and identified some mistakes that were corrected by the authors. This feedback highlights the potential of the method and the authors' willingness to address issues raised during the review process. However, given the detailed analysis and the borderline nature of this submission, I lean towards a borderline reject recommendation. The primary concerns regarding the method's assumptions, its ability to generalize under various conditions, and the restrictiveness of certain constraints need more thorough addressing.

In conclusion, while this paper proposes an interesting approach to graph OOD generalization, it falls short in addressing critical concerns raised by the reviewers. To strengthen their contribution, the authors should comprehensively address these concerns, particularly by providing more robust analysis or experiments that demonstrate the method's ability to handle complex cases and relax some of the restrictive assumptions and constraints. This would enhance the method's applicability and robustness, making it a more significant contribution to the field. Ultimately, this submission highlights important challenges in achieving graph OOD generalization and underscores the need for future work to develop more robust methods that can handle a wide range of scenarios with fewer key assumptions about the data distribution.

**Additional Comments On Reviewer Discussion:**

There was a good discussion and reviewers were engaged. See my comments above.

---

### Decision · Program_Chairs · 2025-01-22

Reject